# Review on Alternative Route to Acrolein through Oxidative Coupling of Alcohols

Vincent Folliard [1,*] , Jacopo de Tommaso [2] and Jean-Luc Dubois [2,*]

[1] Institut de recherches sur la catalyse et l'environnement de Lyon (IRCELYON), Univ Lyon, Université Claude Bernard Lyon1, CNRS, IRCELYON, 2 Avenue Albert Einstein CEDEX, 69626 Villeurbanne, France

[2] Arkema France, Corporate R&D, 420 Rue d'Estienne d'Orves, 92705 Colombes, France; jacopo.de-tommaso@arkema.com

\* Correspondence: vincent.folliard@ircelyon.univ-lyon1.fr (V.F.); jean-luc.dubois@arkema.com (J.-L.D.); Tel.: +33-472-398-511 (J.-L.D.)

**Abstract:** Oxidative coupling of alcohols using methanol and ethanol, which can both be made renewable, is an attractive route to produce acrolein (propenaldehyde) in a single-step process. Currently acrolein is produced by direct oxidation of fossil propylene, and catalytic double dehydration of glycerol has been also investigated up to pilot scale. Although glycerol is an attractive feedstock, it suffers of several drawbacks. Addressing the limitations of both routes, the oxidative coupling of alcohols combines an exothermic oxidation and cross-aldolization. The best performing catalysts so far combine redox and acid/base sites. Reviewing the academic and patent literature, the present paper also addresses the economic analysis, to highlight the potential of this reaction at a yield from 70%, and at two different plant scales. The analysis has been made to guide further research, with the remaining technical problems to solve. Improved selectivity contributing to reduce the amount of equipment and the investment cost should be the prime target.

**Keywords:** methanol; ethanol; acrolein; oxidative coupling of alcohols; catalyst; economic; acid/base; plant; process

## 1. Introduction

Acrolein, also known as propenaldehyde, is a synthetic chemical currently produced by propylene selective oxidation (Reaction B, Scheme 1). It is used for the production of fine chemicals such flavor and fragrances and pharmaceutical compounds, including antibiotics and antimicrobials [1–4]. It is also used in larger quantities to produce methionine (an essential amino acid), and as a non-isolated intermediate to acrylic acid. Therefore, it is produced in several million tons annually.

$$HCHO + CH_3\text{-}CHO \rightarrow CH_2\text{=}CH\text{-}CHO + H_2O \qquad (A)$$

$$CH_2\text{=}CH\text{-}CH_3 + O_2 \rightarrow CH_2\text{=}CH\text{-}CHO + H_2O \qquad (B)$$

$$CH_2(OH)\text{-}CH(OH)\text{-}CH_2(OH) \rightarrow CH_2\text{=}CH\text{-}CHO + 2H_2O \qquad (C)$$

$$CH_3OH + CH_3CH_2OH + O_2 \rightarrow CH_2\text{-}CH\text{-}CHO + 3H_2O \qquad (D)$$

**Scheme 1.** Acrolein production routes: (A) aldol condensation; (B) propylene oxidation (state of the art); (C) glycerol dehydration; (D) oxidative coupling of alcohols.

It has been produced in the past by the condensation of formaldehyde and acetaldehyde [1,5] (Reaction A, Scheme 1). More recently, there has been a lot of interest in the conversion of glycerol, a co-product of biodiesel and oleochemistry, through double intramolecular dehydration [6] (Reaction C, Scheme 1). This was known as a first route to

have access to a renewable source of acrolein, and eventually to acrylic acid, acrylonitrile [7] and several other chemicals using acrolein as a platform molecule. However, there are important challenges with the glycerol route that we will detail below.

There is a need to find alternative processes, which would further reduce the production cost, the carbon footprint or global warming potential. The oxydehydration of 1,3-propanediol, which would lead to allylic alcohol as an intermediate, further oxidized to acrolein could be an option [8], if 1,3-propanediol was not an expensive substrate (about 2.25 US $/kg in 2015–2016, when crude oil was at similar prices as nowadays). The condensation of formaldehyde and acetaldehyde has a proven track record [9–16], but requires to build a methanol oxidation plant and an acetaldehyde plant, in addition to the condensation plant. So, it is a CAPEX (capital expenditure) intensive solution.

Formaldehyde can be produced through two processes: the oxidation of methanol with iron-molybdate catalysts, in multi-tubular reactors, or with silver catalysts through oxydehydrogenation, in shallow bed reactors operated at high temperature [17–19]. Acetaldehyde is produced by ethylene oxidation, with Pd-Cu catalysts [20,21], but it can be also produced through ethanol oxidation or oxydehydrogenation much like methanol [22]. Therefore, we proposed to combine all the reactions in a single step that we called an oxidative coupling of alcohols [23,24] (Reaction D, Scheme 1). In that process, methanol and ethanol are cofed to an oxidation reactor, using a single catalyst, catalyst mixtures or dual beds.

## 2. State of the Art and Review

### 2.1. Curent Route from Fossil Resources: Propylene

Propylene oxidation to acrolein is done in multi-tubular fixed-bed reactors [25]. In this process, a molybdenum-bismuth-iron oxide catalyst (also containing W, P, K, Co, Ni, etc.) in various forms (extrudates, spheres, etc.) is used. It can be supported or bulk. The reactor is operated under a slight over pressure, as the gases have to flow through and get to the downstream purification unit. The catalyst lifetime can be over a dozen years for acrolein production, or shorter when the targeted product is acrylic acid. Indeed, in that case there are two tandem reactors with two different catalysts. Acrolein is directly fed to the second reactor without an isolation step. Therefore, the first reactor is operated in harsher conditions to try to maximize the yield, and that shortens the catalyst life. When acrolein is the targeted product, the reactor temperature is lower, as the goal in this case is to avoid over oxidation. Unconverted propylene can be recycled back to the reactor after capture of the "condensable" products like acrolein. At this stage gaseous products include $CO$, $CO_2$, $O_2$, $N_2$, argon, propylene, propane (which contaminated the propylene stream) and some steam. Since the oxidation reactors are to be operated outside of the flammable zone of the air–propylene–steam domain, diluted air is necessary. Therefore, the gas mix from the reactor exit is a perfect choice as dilution gas because most of the oxygen has been consumed, and the gas still contains some valuable propylene when operated at partial conversion. Unfortunately, the gas also contains some CO that would have adverse effects: for example, oxidizing over the catalyst again, generating excessive heat and a high hot spot and in reducing the catalyst surface. This type of technology therefore requires perfect temperature control. A smaller tube diameter would offer a better control and a higher selectivity, but at the expense of a higher pressure drop (and so variable cost) or a higher reactor capital cost (fixed cost).

Of course, another challenge with this process is that it relies on a propylene sourcing, which is mostly fossil based today, and which is available only from large petrochemical hubs.

### 2.2. The New Route from Renewable Resources: Glycerol

About 20 years ago, alternative routes to produce acrolein from renewable carbon sources were more deeply investigated [26–28]. Glycerol is a by-product of the oleochemical industry and of the biodiesel industry. Unlike what is often reported, it is not a waste because there are multiple applications for it. However, early in the 2000s we were ex-

pecting to see a lot of biodiesel (known as fatty acid methyl esters, or FAMEs) to come on the market all over the world. Biodiesel production from vegetable oils or animal fats is a rather straightforward process, for which rapeseed oil is very appropriate. Depending on weather conditions, other oils and fats can also be used such as palm oil in summer fuel specifications. The glycerol production is always about 10 wt % of the biodiesel production, and in oleochemical industry it is also about 10 wt % of the oils processed. Thus, it was expected to become a major source of feedstock. There are several qualities of glycerol on the market, including crude and refined glycerin. Crude glycerin is a mix of glycerol, water, salts and other organics (non-glycerinic organic matters), and it may also contain some methanol from the biodiesel process. It has to be purified, through a thin film evaporation process or short path distillation, in order to collect a high-purity, refined glycerin. Glycerin containing a small amount of water is a highly viscous material at room temperature and could freeze in winter conditions. This is not the most appropriate form in which to store glycerol. When it contains some amount of water, it becomes an anti-freeze solution, which is much less viscous. Crude glycerol, which can be up to 80 wt % glycerol, is an appropriate form for long-term storage. However, for small consumers a refined glycerin is more appropriate.

Glycerol intra-molecular double dehydration leads to acrolein in yields that are easily above 70 mol %, and sometimes above 80 mol %. The reaction requires an acidic catalyst, such as $W/Al_2O_3$ or preferably $W/TiO_2$ [29]. There have been numerous catalysts reported for that reaction, but very few reached the pilot plant stage. The main issue in this reaction is the steep deactivation of the catalyst due to coke formation, when the reaction is operated in absence of oxygen. With coke build up, the catalyst deactivates in a matter of a few hours and sometimes less (when the reaction is carried in industrially relevant conditions, i.e., with high glycerol partial pressure, high federate, low contact time and 10 0% conversion). Very early, it was proposed to cofeed oxygen (at low partial pressure) in the form of air or diluted air, together with glycerol [27]. These conditions lead to a longer reaction cycle, but also significantly less impurities such as acetone or propanaldehyde, for example. These impurities are made through hydrogen transfer reactions, from the carbon deposit that evolves towards coke to acrolein or other intermediates. These impurities are very important since they are difficult or impossible to remove from acrolein with simple purification technologies and would hamper the potential applications. The amount of propanaldehyde observed in glycerol-derived acrolein is often higher than what is detected in propylene-based acrolein. Although not reported in publications, it can be present above 1000 ppm. Both acrolein and propanaldehyde have the same boiling point, so they cannot be separated by the usual distillation.

With the best catalysts, at high reagent loads, cycles of several days have been obtained. The coke which is formed on the catalyst has often be suspected to be detrimental to the reaction since the deactivated catalyst has a high coke content. However, it was shown by Dalil et al. [30,31] that in the first hours of the reaction, a virgin catalyst builds up coke, and that at the same time the acrolein selectivity is increasing. Some fresh catalysts were purposely contaminated with aromatic coke precursors; it was shown that the catalysts had a better initial selectivity, and that the side products made by hydrogen transfer depend on the hydrogen content of the coke precursors [31].

Several reactor designs have been reported for this reaction, which can be carried out in fixed-bed, fluid-bed, circulating fluid-bed, etc. Since the catalyst deactivates from increased coke build-up, it has to be regenerated periodically. The acceptable period is an important criterion for the choice of technology. If the plant has to be operated continuously, 24/7 (24 h/day and 7 days/week), then a fluid bed with internal or external regenerator, or a circulating fluid bed is appropriate. The cycle of the catalyst (from deactivation to next deactivation) is only going to impact the loss of carbon due to the coke formation. Multiple fixed-bed reactor can also be appropriate, and the sequence of reaction–regeneration–purges has to be well calibrated to have a continuous production. However, the regeneration can be highly exothermic due to the high coke formation (several wt %),

and this requires special design of fixed-bed reactors (more expensive than simple adiabatic reactors, which could be sufficient for the reaction alone). If the plant is to be operated on a weekly basis, which might be appropriate for small producers, one could have cycles of 4 days production and 3 days regeneration. In that configuration, the catalyst must have a slow deactivation, and time would be available for a smooth regeneration over the weekends for example. These considerations illustrate the impact of the business plan on the reactor design and catalyst selection.

The reaction is an intramolecular dehydration of glycerol, but intermolecular dehydration can also take place. This is most probably the route that leads to the heavy products which are going to end up as coke. It was quickly identified that the acrolein formation works better when the glycerol dehydration is done in the presence of water. This is explained by the impact of the water partial pressure on the kinetics of the two main reactions: intra- and intermolecular dehydration. The reverse reaction of the intermolecular dehydration is more favored in the presence of steam. Thus, hydrous glycerin is not only preferable for winter storage conditions, or to improve the pumpability (reduce the viscosity) of the glycerin, it is also necessary to improve the selectivity of the reaction and to extend the catalyst useful life. However, the energy consumption of the process will be strongly impacted as water has also to be evaporated.

A major interest in the "glycerol" route to acrolein is then the ubiquitous character of glycerol, which is appropriate for small consumers of acrolein. The feedstock is not only available nearly everywhere, but also easy to store and transport unlike propylene. It is also safer to store glycerol and produce acrolein on demand than to transport and store acrolein. The two major industrial accidents related to acrolein, in Taft (USA) and Pierre-Bénite (France), were related to, respectively, storage and transport, fortunately without fatalities [32].

Unfortunately, the market of biodiesel did not develop as initially expected, mainly because of the numerous changes in the EU biofuels regulations, and today the volumes available in Europe do not allow considering a large-scale plant. Assuming a yield of 70 mol %, glycerol dehydration would correspond to 42 wt % yield, and in other words 1 ton of acrolein requires 2.3 tons of glycerol, or an equivalence of 23 tons of biodiesel. Therefore, a "small" acrolein plant of 10,000 tons would consume nearly all the glycerol production from a large 250,000 tons biodiesel plant. Thus, the sourcing of glycerol is another major challenge in the production of biobased acrolein.

### 2.3. The New Route from Mixed Alcohols: Oxidative Coupling of Alcohols

Still looking for an alternative process that would satisfy technical, environmental and economic criteria, some of the authors investigated the "oxidative coupling of alcohols". In order to minimize the capital cost, i.e., the number of processing units, we need to have a single-step reaction, operated at low contact time. The "old" condensation of formaldehyde and acetaldehyde would require to have a reactor for methanol oxidation/oxydehydrogenation, another one for acetaldehyde production and a last one for the aldol condensation, and multiple purification steps. Instead, the goal with the new reaction system is to have a single reactor, in which methanol and ethanol are cofed with an oxygen source. The catalyst has to be able to oxidize both methanol and ethanol; and iron-molybdate is the formulation of choice, since it is so far the best for methanol oxidation and has been tested previously in ethanol oxidation [22]. It is also a formulation of choice since acrolein was shown to be stable in the presence of iron-molybdate at low contact time [33]. Low contact times mean also that a low catalyst volume and, thus, a small reactor are sufficient, thereby reducing the capital cost. The reaction is an exothermic oxidation, carried at high temperature (above 200 °C), so with proper reactor design the reaction energy can be recovered as high-pressure steam, which will be used in down-stream process steps to purify the products by distillation and for energy production. That way the process can be energy sufficient, at a reasonable yield like for the propylene oxidation process.

Ethanol is a renewable carbon source, as it is produced by fermentation of various sugars. Sustainable methanol sources are becoming increasingly available. It has been produced from glycerol (by BioMCN for example), through $CO_2$ hydrogenation [34], and more recently it is going to be produced from biogas by a direct oxidation process [35]. In addition, both feedstock are widely available (not yet everywhere for renewable methanol, but it will come), easy to store and to transport. Although some safety is still required, the constraints are not as restrictive as for acrolein. This new route has very favorable grounds compared to the alternative feedstocks reviewed above.

Experimental work to validate the proof of concept has been carried out over several years by Dubois, Auroux, Capron and co-workers [23,36–42]. As explained above, the ultimate goal in this reaction is to have a single reactor to minimize the capital cost (at least not higher than the propylene oxidation process). In that configuration, the catalyst bed can be made of a single catalyst combining both redox and acid/base functions, or it could be made of multiple layers of oxidation and acid/base catalysts. Early on, we discovered that some single catalysts are able to produce acrolein from methanol and ethanol, such as the iron-molybdate catalyst itself, although with a moderate yield of a few percent.

In a single reactor configuration, Borowiec et al. [36,37] studied the production of acrolein by oxidative coupling of methanol and ethanol using modified iron-molybdate catalysts (FeMoOx). In their first study, they synthesized FeMoOx with three different Mo/Fe ratios (1.5, 2.0, 2.5) by adjusting the calcination temperature. Among important results, the authors showed that a calcination temperature of 400 °C was necessary to produce the crystallization of $MoO_3$ and $Fe_2(MoO_4)_3$, which are responsible for the catalytic activity. Among the studied catalysts, the best catalytic activity was displayed by an iron molybdate with a Mo/Fe ratio equal to 2.5 (39% of acrolein at 400 °C) [36].

Subsequently, to enhance acrolein yield, they modified the iron-molybdate catalysts by doping with lanthanum (La) and cerium (Ce). Those solids generate 42% of acrolein yield with FeMoCe2.5 and 40.5% acrolein with FeMoLa2.5. Finally, by using basic mixed oxides such as MgO, CaO or BaO as co-catalysts to FeMoOx, the authors managed to obtain 49% of acrolein, underlining that addition of basicity to the catalyst was beneficial to the acrolein production [37].

To better understand the reaction mechanism, the reaction was also investigated at partial conversion, and using two different reactors, in order to properly tune the redox and acid/base properties, or in a single reactor. In the two-reactors (tandem) configuration, the first reactor, operated at partial conversion, is devoted to the oxidation of the alcohols (methanol + ethanol) to produce formaldehyde and acetaldehyde, respectively, over an iron-molybdate catalyst (FeMoOx). Then, the mixture of aldehydes, remaining alcohols, oxygen, water and secondary products is directed to a second reactor to perform aldolization reaction and produce more acrolein.

Water produced by the reactions has adverse effects on the aldolization reaction. In addition, $CO_2$ can have an adverse effect on the basic sites, and $O_2$ can trigger over-oxidation of products. The methanol-to-ethanol ratio, in experiments, has been either 1 or 2, depending if the goal was to improve the overall yield or to better understand the reaction mechanism.

Because the first step is an oxidation, it is important to know the flammability limits of the considered reagents. Tabulated data are accessible elsewhere for methanol and ethanol, in air/nitrogen mixtures [43], for atmospheric pressure at room temperature. Besides, in [44], the variation of the limits with temperature and inert gas composition are illustrated, as for example the effect of temperature. Upper flammability limit (UFL) is also affected by pressure, which is not too problematic for oxidative coupling of alcohols because the reaction is planned to be performed close to atmospheric pressure. Lower flammability limit (LFL) is not modified too much by pressure and temperature changes [39].

Of course, nothing can be better than measurement of flammability limits. As done for Lilic et al. [39], we used the Le Chatelier rule (Equation (1)) to get a fair approximation of the upper (UFL) flammability limits for alcohol mixtures:

$$1/\text{UFL (mix)} = x_1/\text{UFL (Methanol)} + x_2/\text{UFL(Ethanol)} \tag{1}$$

and a similar equation for the lower flammability limit (LFL). In this equation, $x_1$ and $x_2$ represent the relative concentrations of methanol and ethanol in the alcohol mixture. On the basis of data reported in the publication of Brooks and Crowl [43], the limits were determined for multiple oxygen/nitrogen compositions, and the graph for a 50 %/50 % methanol/ethanol mole ratio was calculated and reported in Figure 1. On the basis of this diagram, a feed composition range outside of the flammability area can be determined to operate the unit safely. There are advantages to have a higher alcohols partial pressure, as the unit productivity could be increased. The oxygen partial pressure needs to be higher than the stoichiometry to compensate for the non-selective oxidations to CO and $CO_2$, and to make sure that enough oxygen remains at the end of the oxidation reactor to avoid catalyst over reduction. The reactor also has to be able to manage the heat of the reaction, and avoid runaway conditions, in which the temperature increase would become uncontrolled. Fluid bed reactors are probably not the most appropriate since acrolein is not so stable, but the technology is appropriate to feed reagents slightly in the flammability limits since oxidant and alcohols would be fed in different locations. However, a circulating fluid bed reactor operated in a redox mode, with re-oxidation of the catalyst in a regenerator, and low oxygen partial pressure in presence of acrolein, could be appropriate. A multi-tubular Fixed-bed reactor, which is also the technology used for methanol oxidation to formaldehyde and propylene oxidation to acrolein, has been considered for oxidative coupling of alcohols.

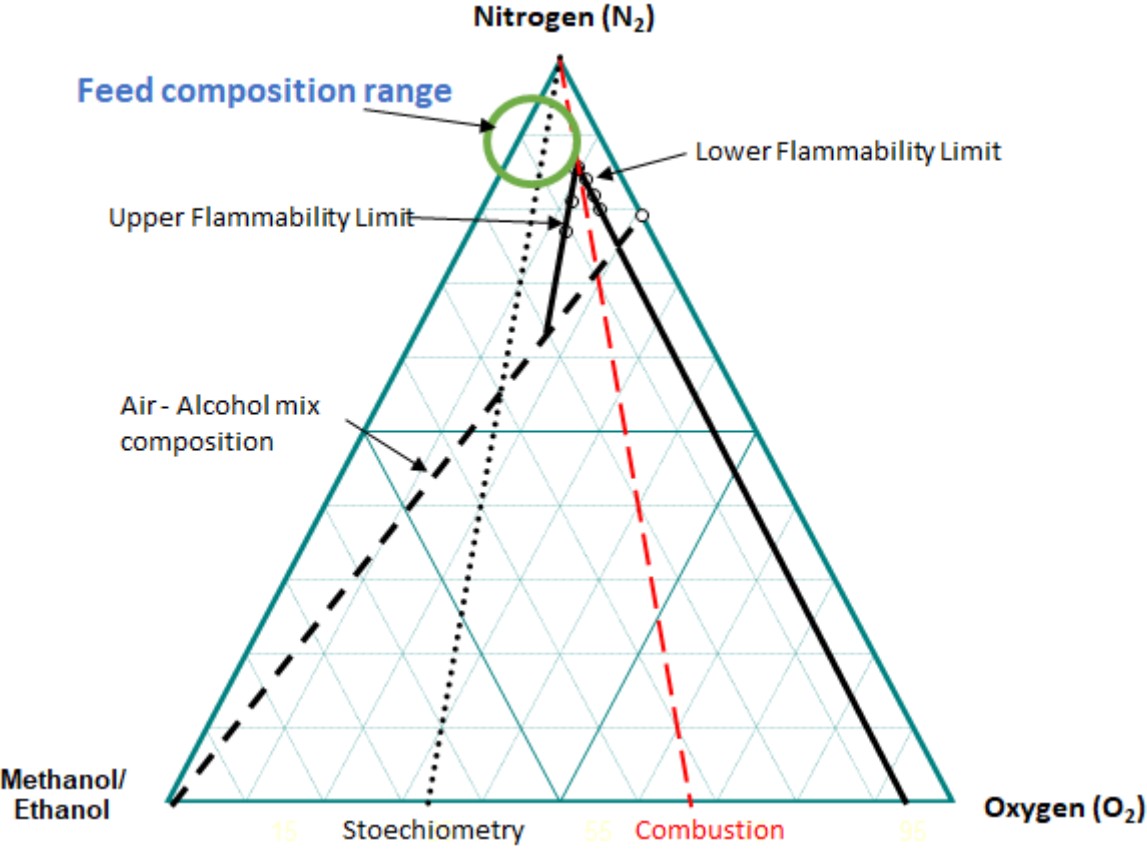

**Figure 1.** Flammability limits diagram at 25 °C for methanol and ethanol mix = 50%:50% molar ratio.

Based on the above considerations, the alcohol content in the feed stream should be around 10%, and the oxygen content should be below 10%. The flammability diagram is also important to understand the conditions to be used for reactor start-up conditions [45]. Starting from air composition, in which the catalyst was loaded, inert gas has to be added to go around the flammable zone. First, a small amount of alcohols is added, and exhausted gas is recycled to dilute the air sufficiently. Then, alcohol partial pressure can be increased together with a reduction in the diluent feed.

- Acid/base balance

The second step is a cross-aldolization reaction catalyzed by both acids and bases with two different reaction mechanisms.

Cross-condensation/aldolization of formaldehyde and acetaldehyde has already been extensively studied [9–16,46–49]. Many catalysts have been used for this reaction such as mixed oxides, phosphates, zeolites or clays. Nonetheless, despite the large number of studies, the mechanism of cross-condensation of acetaldehyde and formaldehyde is still unclear.

As early as the 1960s, multiple studies on cross-condensation of aldehydes were carried out by Malinowski et al. In their first study [46], the authors prepared basic catalysts by saturating silica gel with an aqueous solution of sodium hydroxide (NaOH). They observed good yields at temperatures between 200 and 350 °C with only 2% of co-products. The authors completed this work by a kinetic investigation, which allowed to establish that the reaction was accelerated by the presence of sodium-containing centers, probably Si–O–Na, and that the different values of the reaction rate constants resulted from the number of these centers.

Later, Malinowski et al. studied the same reaction over silica-alumina catalysts soaked with aqueous hydrofluoric acid [49]. During experimental tests, in addition to acrolein, the authors observed acetone and propionaldehyde among the obtained products.

The authors observed that acrolein was the main product up to 450 °C, but that its yield drastically decreased above 350 °C. Acetone increased from 350 °C to reach a maximum at 400 °C. Above 400 °C, propionaldehyde appeared until a maximum at 450 °C.

Following this observation, the authors proposed an ionic mechanism, where donor and acceptor centers are responsible for the acrolein production.

They also proposed a radical mechanism to produce acetone, even if to our opinion it is more probable that acetic acid, which comes from the acetaldehyde, transforms itself in acetate, which produces acetone over basic catalysts (Piria reaction mechanism).

Finally, the authors proposed that propionaldehyde was produced by the reaction of methanol, coming from formaldehyde disproportionation, and acrolein. Nonetheless, it also possible that hydrogen transfer reaction happens directly on acrolein to produce propionaldehyde.

In the case of oxidative coupling of alcohols, those by-products produced in relatively high quantity should be avoided since the goal is to work at lower temperatures than 350 °C.

A study performed in the 1990s, in gas phase over oxides supported on silica gel [10], showed that addition of an alkaline base or an oxide such as amphoteric $Al_2O_3$ or acidic $V_2O_5$ increased the catalytic activity. In a second study performed over metal oxide and phosphate catalysts, the same author reported that addition of acidic oxide to the amphoteric compound was beneficial to the acrolein production [9]. Then, it appears that both basic and acidic sites play a role in the reaction.

Later, vapor phase aldol condensation of formaldehyde and acetaldehyde was widely studied by a Romanian team [11–16,48].

In 1993, Dumitriu et al. studied acrolein production by cross-aldolization of formaldehyde and acetaldehyde, over oxides such as $MoO_3$, $MgO$, $ZnO$, $B_2O_3$ or $P_2O_5$ deposited on silica, alumina, or zeolites, Y-faujasite and ZSM-5 [12]. They observed that addition of metallic oxides over HZSM-5 increased acrolein yields; however, the inverse effect has been noticed for $B_2O_3$ or $P_2O_5$ deposited on silica. The best results were seen with MgO

deposited on HZSM-5. The authors explained this behavior by the possible cooperation between basic centers of MgO and acidic sites of zeolites HZSM-5.

In another work [14], they investigated aldol condensation over hydrotalcite-like compounds, and compared it to magnesia (MgO), alumina ($Al_2O_3$), and mixed oxides (MgO-$Al_2O_3$). The authors reported that basic magnesia seemed to favor acrolein production, while amphoteric alumina favored self-aldolization of acetaldehyde. From their obtained results, the authors proposed a reaction mechanism where cooperation between basic and acidic sites takes place. Indeed, acetaldehyde would adsorb on basic sites and formaldehyde on weak acidic sites. Nonetheless, the authors underlined in another study that both basic and acidic sites were able to activate the two aldehydes, but that basic properties were governing the reaction network [15].

For the oxidative coupling of alcohols, where aldol condensation of formaldehyde and acetaldehyde is the consecutive step, influence of acid/base properties of catalysts has been studied by Lilic et al. and Folliard et al. [38–42]. In these experiments, the goal was to better understand the reaction mechanism, not to get the best results. Thus, the reaction conditions were selected to be at partial conversion, so that some reagents were still present during the aldolization reaction. Thanks to adsorption microcalorimetry, acidic and basic properties of each catalyst were determined, and scales of acidic and basic strength were created. These scales are available in Figure 2 and show the variety of studied catalysts such as basic oxides supported on silica, amphoteric hydrotalcites, spinels and acidic heteropolyacids.

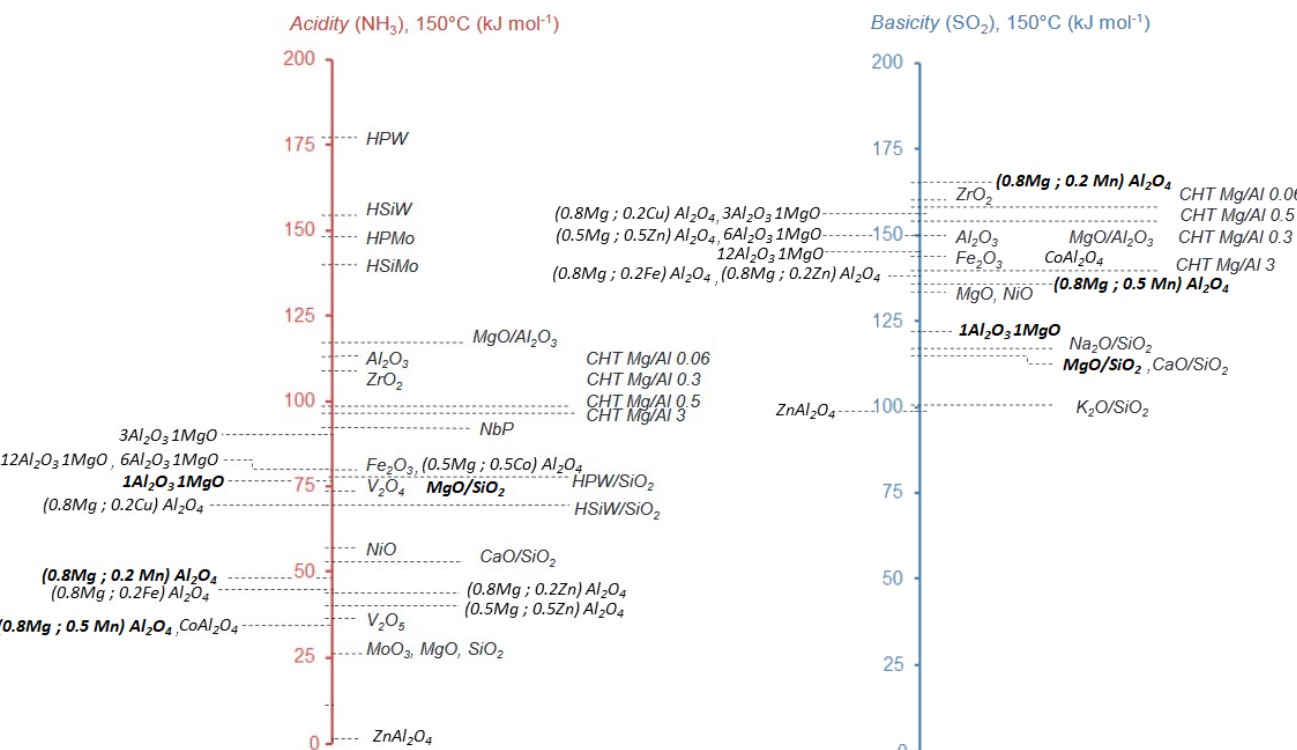

**Figure 2.** Scales of acidic and basic strength based on the average heat of probe molecule adsorption measured at the plateau of the differential heat curve, or at half coverage when no plateau is observed. See references [39,40] for more details on experimental conditions. Bold: best catalytic systems.

Concerning the catalytic activity, over oxides supported on silica, the best result was displayed by MgO/$SiO_2$ with 35% acrolein yield (at 320 °C and 5000 $h^{-1}$ Gas Hourly Space Velocity—GHSV) followed by NaO/$SiO_2$ with 25% acrolein production (at 340 °C and 5000 $h^{-1}$ GHSV). In this work, the authors calculated the ratio of the number of strong

basic to strong acidic sites and correlated it to the acrolein yield. It appears that an excess of strong basic sites was detrimental to the acrolein production.

A second study performed over basic, amphoteric and acidic catalysts led to the conclusion that aldol condensation of acetaldehyde and formaldehyde to acrolein in oxidizing condition was taking place on both acidic and basic sites [38]. The authors underlined that basic sites allowed to increase acrolein yield but also increased $CO + CO_2$ production, while on acidic sites neither acrolein nor COx yield increased. Finally, the authors proposed that coexistence of strong acidic and strong basic sites in similar amounts was the best configuration to optimize the acrolein yield.

In 2020, Folliard et al. studied the same reaction in the same conditions using spinel catalysts [40]. In a first series of spinels, where $Al_2O_3/MgO$ ratios were varying, they obtained the best acrolein production with $1Al_2O_3,1MgO$ with 27% acrolein (at 285 °C and GHSV 5000 $h^{-1}$). Calculation of the number of strong basic to strong acidic sites ratio confirmed the previous results obtained by Lilic et al. ([38,39]), underlining the detrimental effect of an excess of strong basicity compared to strong acidity [40].

In another study [42], the same team studied the influence of partial or total substitution of magnesium in spinel by transition metals (Co, Cu, Fe, Mn, Zn). In this case, the best catalytic activity was displayed by $(0.8Mg; 0.2Mn)Al_2O_4$ with 31% acrolein yield (at 285 °C and GHSV 5000 $h^{-1}$). However, this time, after calculation of the ratio of number of strong basic sites to strong acidic sites, they did not succeed to correlate it to acrolein production, indicating that not only the acid/base properties of the fresh catalyst, but also other parameters were influencing the acrolein production. Noticeably, the authors have been able to correlate the acrolein production to increasing radius of substitution metal, suggesting that electronic parameters, such as electronic density around cationic center, could have an influence on acrolein production.

With the catalysts investigated so far in oxidative coupling of alcohols, the reaction needs both acid and basic active sites on the fresh catalyst. A first major difference with prior literature is that the reaction is carried out in the presence of oxygen, and a second major difference is that the reaction medium is also composed of alcohols, water, COx and at least two aldehydes. Several publications focused on the self-aldolization of acetaldehyde or butyraldehyde, in the absence of water and oxygen [50–54]. When cross-aldolization of acetaldehyde and formaldehyde was investigated, the reaction medium contains water and some methanol because of the formalin solutions that have been used as formaldehyde source. Other cross- and self-aldolizations are also done in liquid phase with sodium hydroxide or amines as catalysts [55]. In strong alkaline conditions, dehydrogenation/oxidation reaction can occur, and that would contribute to lower the yield of the reaction. In our case, the conditions we investigated are rather unconventional.

- Mechanism

The reaction is supposed to be an oxidation coupled with an aldolization. Though surprisingly, although both formaldehyde and acetaldehyde are coproduced, the self-aldolization product of acetaldehyde (crotonaldehyde) was detected only in minute quantities. The first thought was that this product was not thermodynamically favored as supported by thermodynamic calculations [38]. However, the difference between self-aldolization and cross-aldolization is not very large either. Experiments were done, co-feeding crotonaldehyde in the process conditions, but no acetaldehyde was observed. Thus, this suggested that the absence of crotonaldehyde is rather kinetically controlled.

Folliard et al. carried out adsorption micro-calorimetry experiments, where acetaldehyde and formaldehyde were successively sent on acid/base catalysts [40]. The experiments revealed that formaldehyde is more strongly adsorbed on the surface than acetaldehyde. Which means that, in the reaction conditions, adsorbed acetaldehyde moieties are isolated in a large pool of formaldehyde. Therefore, their only reaction opportunity is to react with a neighboring formaldehyde and desorb as acrolein. To preserve a high selectivity of the reaction, it is therefore important to keep a high concentration of adsorbed formaldehyde.

- Catalyst at equilibrium

Catalysts have been extensively characterized before reaction. After an initial phase, the catalysts slowly deactivate. Like in the case of the glycerol dehydration, the catalyst is conditioned or equilibrated under the reaction conditions. Folliard et al. determined the acidic and basic properties by adsorption micro-calorimetry over spinel catalysts with varying $Al_2O_3$/MgO ratios before and after 5 h reaction [40]. Figure 3 displays the ratio of strong basic to strong acidic sites versus duration of reaction.

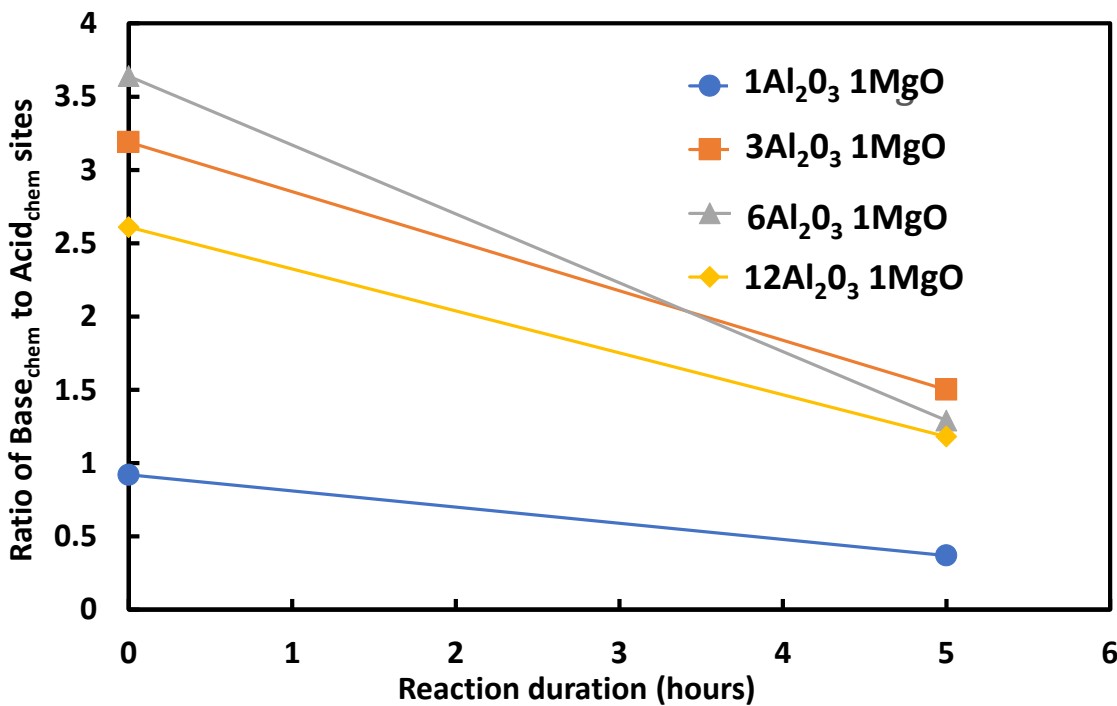

**Figure 3.** Evolution of ratio of strong basic to strong acidic sites after 0 and 5 h of reaction for spinel catalysts with varying alumina to magnesia ratios.

After only 5 h of reaction, the ratio of strong basic to strong acidic sites strongly declined, probably because of coke or carbonate deposition which poisoned basic sites, showing equilibrated catalysts with a significantly different acid/base balance compared to the virgin catalyst. Nonetheless, catalysts were still active and selective for the targeted reaction.

Figure 4 gathers some results obtained by Lilic et al. and Folliard et al. [38–42], namely acrolein production (around 270 °C and 5000 h$^{-1}$) versus the calculated ratio of number of strong basic to strong acidic sites, obtained over fresh catalysts. At a first glance, acid/base properties have a strong impact on the acrolein production over supported samples (green zone), and surface acid/base properties are easier to tune compared to bulk catalysts such as xMgO yAl$_2$O$_3$ spinels (grey zone) or hydrotalcites (blue zone). In addition, the figure points out that it would be preferable to target a ratio of strong basic to acidic sites close to 1 to enhance the acrolein yield. The ideal catalyst properties should not be directly derived from the fresh catalyst, but more efforts should be devoted to characterization of working catalysts, either operando or off-line, to better determine the key properties needed for the reaction. The acid/base balance depends also on the time on stream.

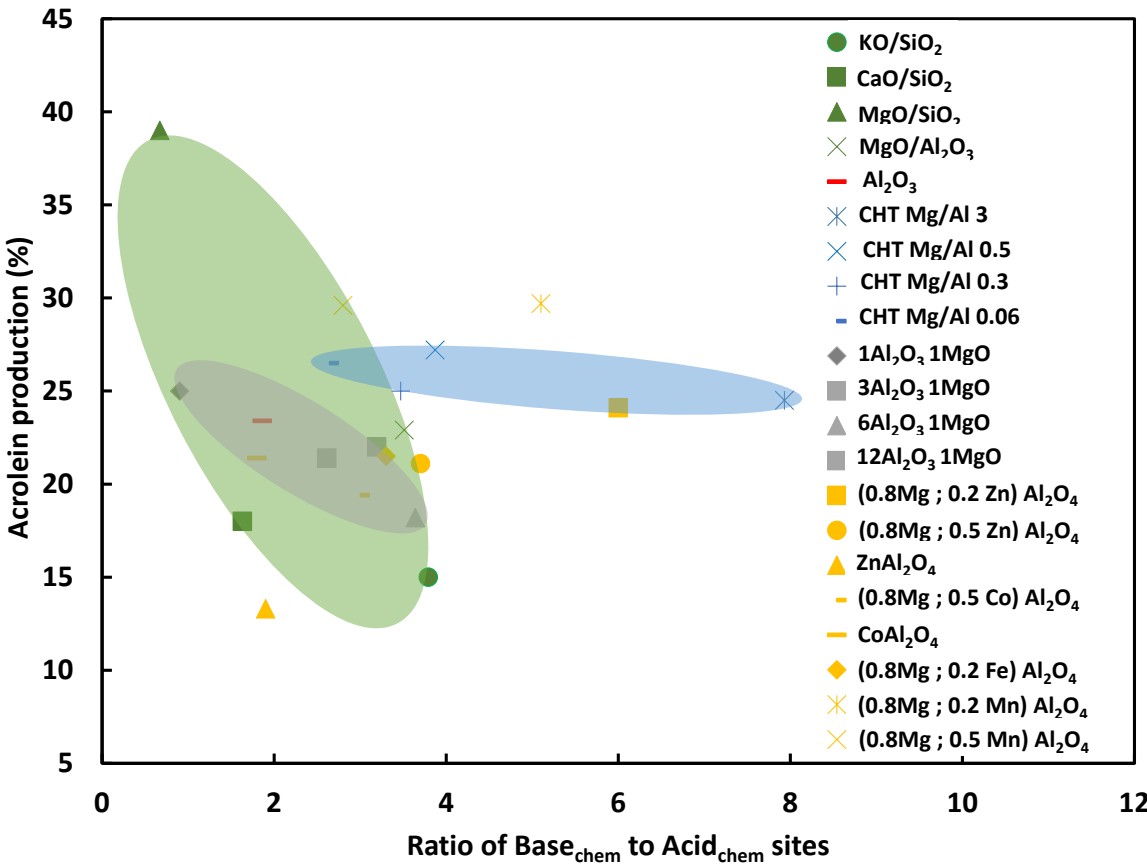

**Figure 4.** Ratio of strong basic to strong acidic sites obtained for fresh catalysts versus acrolein production determined by Lilic et al. and Folliard et al. [38–42].

- Reaction set-up and considerations for the Industrial plant

At a first glance, the commercial plant would look like a classical propylene oxidation plant and/or a methanol oxidation plant, which means centered on a multi-tubular fixed bed reactor. We could have considered a fluid bed reactor, which is a perfect technology to control the temperature of the bed, and which allows to work partially in the flammable zone. This technology is used for propylene ammoxidation to acrylonitrile and for n-butane oxidation to maleic anhydride for example. However, it is not (yet) used for propylene oxidation to acrolein or acrylic acid, or for methanol oxidation to formaldehyde. Several years ago, the circulating fluid bed was investigated for propane and propylene oxidation using a redox cycle, but it has not yet been implemented. One of the reasons why a fluid bed is not as appropriate for acrolein as for acrylonitrile, although the catalysts formulations are very similar, is that acrylonitrile is much more stable at high temperature than acrolein. In a fluid bed, there is a large gas volume above the catalyst bed, in which the product remains under high temperature. In a fixed-bed reactor, as soon as the product leaves the catalyst bed it is quenched to lower temperature, and this makes this technology more appropriate for heat-sensitive products.

The major difference with an acrylic acid plant is that a single reactor is expected. A preliminary design of the plant is illustrated in Figure 5. Air is compressed to about 2–3 bars, and gas flows on two separate methanol and ethanol feeding tanks to get the appropriate vapor pressure of the 2 reagents. The gas is then preheated and enters the multi-tubular reactor. The temperature is controlled by circulation of a molten salt or oil (heat transfer fluid), which generates high-pressure steam since the reaction is carried above 200 °C. The residence time in the reactor is not longer than for propylene oxidation (it is in fact about half shorter in the experiments that have been done so far). The gases leaving the reactor are

quenched on a heat exchanger, which also recovers heat, and enter an absorption column. Uncondensable gases leave the column from the top. The stream is mostly constituted of nitrogen, which cannot react, some remaining oxygen—otherwise the catalyst would get reduced and would deactivate—CO and $CO_2$ as oxidation products, and some argon which is present in the air. Part of the gas is purged to a catalytic incinerator, and part of the gas is recirculated to the entrance of the plant, before the alcohols evaporators, to have diluted air with the right oxygen partial pressure to avoid the flammable zone of the gas mixture, as illustrated with the flammability diagram.

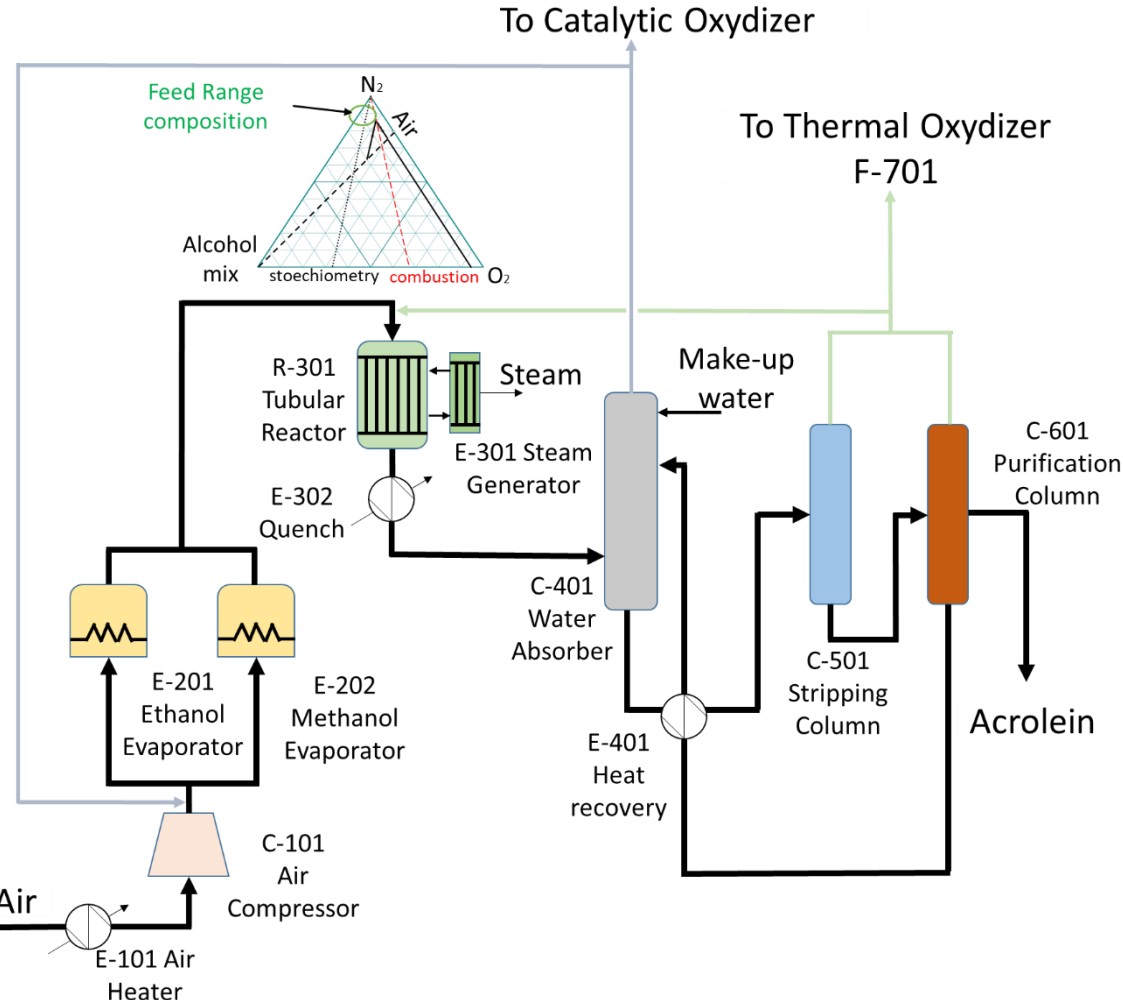

**Figure 5.** Proposed process flow diagram of the new "Oxidative Coupling of Alcohols" route.

The acrolein-rich liquid, which is recovered from the absorption column, follows the same purification train as for the propylene oxidation or glycerol dehydration processes [1,3,4]. However, we expect that the acetaldehyde and formaldehyde concentrations would be higher. The two co-products are partly recovered and returned to the reactor in order to improve the yields.

Another part of the co-products will be directed to the thermal oxidizer, where they will be combusted to generate an extra amount of energy. Because of the exothermic nature of the reaction and because of the heat recovery from the combustion of the by-products, there is enough energy supplied by the process to satisfy the energy demand of the purification and evaporation sections. The process can ideally by divided into seven main steps: 101 (air compression); 201 (reagent evaporation); 301 (reaction), 401 (absorption), 501 (stripping), 601 (purification) and 701 (thermal oxidation).

### 3. Life Cycle and Economic Analysis

When both methanol and ethanol are of renewable origins, all the $CO_2$ that would be produced on the site would be also of renewable origin. Thus, the net fossil $CO_2$ emissions would be null on the site, since the plant is self-sufficient in energy. What needs to be taken into account is the fossil $CO_2$ emission for renewable methanol and ethanol production, and for the electricity generation.

The "Oxidative Coupling of Alcohols" was analyzed through a Monte-Carlo simulation model, in order to determine the conditions that would make this route successful. In order to build this model, a mass balance has to be initiated. We assume that the reaction would proceed with an equimolar ratio of methanol and ethanol. That might be different from the experimental conditions of some academic work, but the latter was targeting a better understanding of the reaction mechanism and was not taking into account the recycling of side products (acetaldehyde and formaldehyde) nor of unconverted alcohols. In addition, we assume a 70% yield as a base case, which means that both reagents are converted at the same yield. To simplify the mass balance, the side product is assumed to be $CO_2$ only. A high amount of energy is generated in the reactor, or outside of the reactor where the side products are incinerated with energy recovery. However, either way it does not change the mass balance (Table 1). The process is supposed to be energy self-sufficient, except for the needs in electricity for which we assumed that the annual cost corresponds to 2% of the raw materials costs.

**Table 1.** Mass balance for big (50,000 tons/year) and small (10,000 tons/year) plants for 70, 80 and 90% yields.

| 70% Yield | Acrolein | Methanol | Ethanol |
|---|---|---|---|
| Big (tons/year) | 50,000 | 40,825 | 58,700 |
| Small (tons/year) | 10,000 | 8165 | 11,740 |
| **80% Yield** | | | |
| Big (tons/year) | 50,000 | 35,720 | 51,360 |
| Small (tons/year) | 10,000 | 7144 | 10,272 |
| **90% Yield** | | | |
| Big (tons/year) | 50,000 | 31,752 | 45,655 |
| Small (tons/year) | 10,000 | 6350 | 9131 |

On the basis of the acrolein market reviewed previously, we considered two possible plant sizes: either 10,000 tons/year or 50,000 tons/year. The small plant size corresponds to a small acrolein consumer that would prefer to become a producer, to avoid transportation and storage of a dangerous chemical compound, but also to be much less dependent on imports and eventual supply difficulties. A different plant size could have been selected, but one should keep in mind that for an even smaller plant, the equivalent acrolein purchased on the market would be slightly more expensive. So, the economic analysis would not be that much affected. The large plant size selected corresponds either to the demand for a methionine plant, or a large demonstrator size for an acrylic acid plant. In the latter case, it is not expected to generate a profit since a demonstrator is designed to validate a technology.

Utilities, here mostly electricity and water, are calculated as 2% of the raw material cost, since the process is self-sufficient in energy. In addition to the feedstock cost, derived from the mass balance, there are other variable costs such as royalties (3% of sales), cost for sales and marketing (10% of sales), and since this is one of the few occasions we can increase it, a comfortable 5% R&D budget calculated on the sales (for a total of 18%).

The number of operators is not much affected by the plant size since the plant has to operate continuously in both cases. We assumed that there would be five shifts of three workers each, at an annual labor cost (LC) of 60,000 US $/year/employee. The supervisors, lab technicians and other managers are proportional to the operator cost. Laboratory

supervision and laboratory charges are both taken at 18% of LC, the plant overhead is 60% of LC, and the administration 20%. The total labor cost was then assumed to be 1,944,000 US $/year.

The fixed cost includes, in addition to the labor cost, the capital cost depreciation on 10 years, but also factored costs for the taxes (2%), insurance (2%), maintenance and repair (2%), operating supplies (1%) and financial interests (2%), with percentages based on the plant capital cost (for a total of 9%).

The capital cost (plant investment) estimate can be done in different ways at this stage. It is a very early cost estimation with very low level of details. Cost estimation methods had been reviewed by Tsagkari et al. for multiple biomass conversion processes [56], and several methods can give fair capital cost estimates with a very low level of plants details. In the present case, we can rely on the so-called Petley method [57] and the Lange method [58]. The first one is based on number of functional units, the plant size, and the maximum pressure and temperature in the process, which are accessible at this stage. The second method is based on the energy loss in a process and is a fair appropriate for a highly exothermic process. It is based on the idea that the capital cost is related to the heat exchanger area.

The original Petley's correlation (from 1988) was updated to 2019, and relocated to France as

$$ISBL\ (2019) = 55,882Q^{0.44}N^{0.486}T_{\max}{}^{0.038}P_{\max}{}^{-0.22}F_{\mathrm{m}}{}^{0.341}\frac{\mathrm{CEPCI}(2019)}{\mathrm{CEPCI}(1988)} \cdot Fl \tag{2}$$

where *ISBL* is the Inside Battery Limit investment, $Q$ is the capacity of acrolein in tons/year, $N$ the number of process steps (7 here), $T_{\max}$ the maximum process temperature in K (573 K), $P_{\max}$ the maximum pressure in bar (2 Bars), and $F_{\mathrm{m}}$ the material construction factor (1.5). The 1988 CAPEX has then to be updated to 2019 and relocated to France (*Fl*). The Chemical Engineering Plant Cost Indexes (CEPCI) update the plant cost dealing with the increased cost of construction over time. In 1988 the CEPCI was 342.5, while in 2019 it was 607.5 [59,60]. Peter et al. estimated that the Outside Battery Limit (OSBL) investment is 25–40% of the ISBL [61], and we chose 40% to be conservative.

Lange correlated the ISBL with the energy losses in the plant, calculated as $\mathrm{LHV}_{(\mathrm{feed\ +\ fuel})} - \mathrm{LHV}_{(\mathrm{product})}$ as

$$ISBL + OSBL\ (2019) = 3.0 \cdot (\text{energy losses }[\mathrm{MW}])^{0.84} \cdot \frac{\mathrm{CEPCI}(2019)}{\mathrm{CEPCI}(1993)} \cdot Fl \tag{3}$$

Again, we updated the investment with the CEPCI of 1993 (343.5) and the relocation factor for France.

A third method, which is also relevant, is by expert judgement. In the present case, knowing the type of process and products, assuming that reaction would use a multi-tubular reactor, like in the propylene oxidation process, and that the rest of the purification would be similar, the capital cost would be close to the current propylene oxidation at the same plant capacity.

The capital cost estimation can be based on process reviews (from expert companies like IHS [62], Nexant or Intratec [63,64]) and on press releases from companies that have built similar plants. In the present case, several brown field acrylic acid plants have been built in the last 10–20 years in different locations, and very few Acrolein plants have been built. These different data are used to make a fair estimate of a capital cost distribution. In Figure 6, we report the investment costs recalculated for the same year (using the CEPCI plant cost index) and the same location (using relocation factors for France).

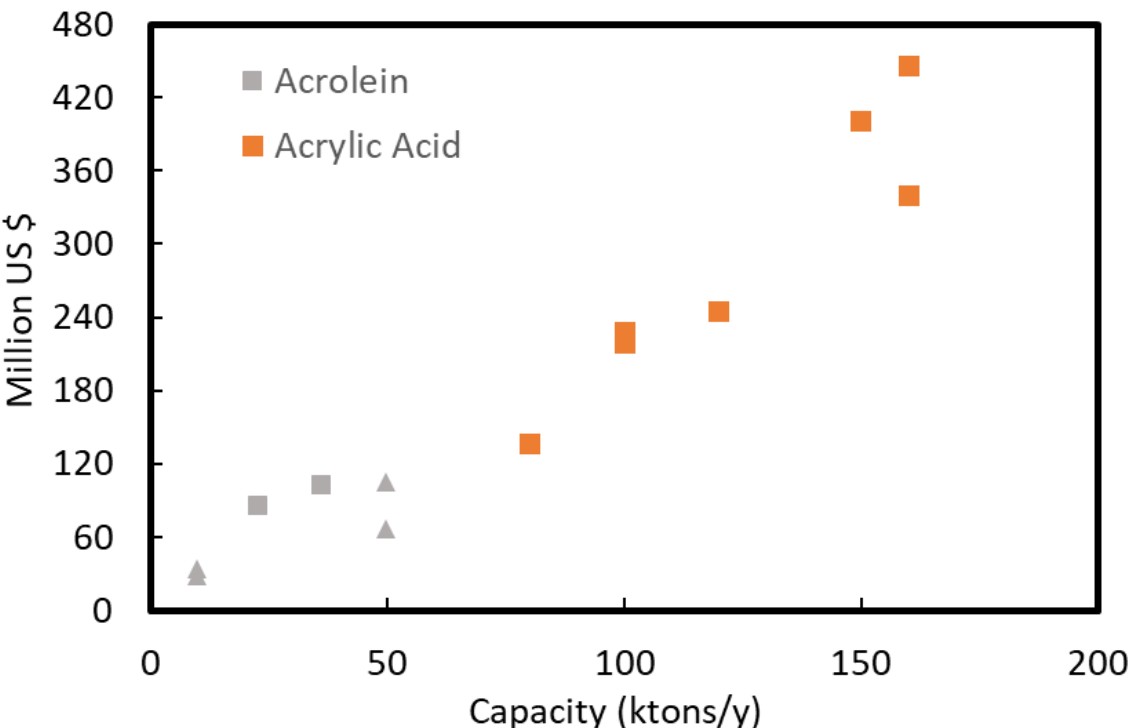

**Figure 6.** Scaled investment (2019, France) versus original plant capacity, with historical data of acrylic acid acrolein production plants (grey squared marks), and early estimation methods (triangular marks) for an acrolein 50 kt/y and 10 kt/y plant at 70% yield.

Data used in Figure 6 are based on construction on new sites (Greenfield) or plant expansion (Brownfield) for acrolein (A) and mostly acrylic acid (AA) plants, Table 2. In addition, data for one acrolein unit process design, two acrylic acid unit designs (e.g., Intratec 350 M$ for 150 ktons/y in 2015 [65]), and estimation based on Lange and Petley equations are also included.

**Table 2.** Historical data for acrylic acid and acrolein plants.

| Company | Capacity (ktons/year) | Investment (M US$) | Year | Project Type | Product |
|---|---|---|---|---|---|
| Nippon Shokubai [66] | 100 | 200 | 2019 | Brownfield | AA |
| Nippon Shokubai [67] | 100 | 195 | 2016 | Greenfield | AA |
| Nippon Shokubai [68] | 80 | 138 | 2011 | Brownfield | AA |
| American Acrylic [69] | 120 | 150 | 1999 | Greenfield | AA |
| BASF [70] | 160 | 200 | 1998 | Greenfield | AA |
| Arkema [71,72] | 36 | 65 | 2003 | Greenfield | A |

For the large BASF/Petronas and the Nippon Shokubai (2016) investments, which include other units such as ester synthesis, we guessed what would be the AA plant cost contribution. Similarly, for the acrolein plant, which is included in a larger investment, we used our expertise to guess the acrolein plant contribution.

For our Monte Carlo simulation (3000 sets of data), we used statistical distributions for cost contributions (except for labor cost). Indeed, with the cost estimation methods used, which can be "Class V" for process engineers, or technology readiness level (TRL) 3–4 for chemists, i.e., very preliminary data, there is a significant uncertainty. To reflect it, we used a probability distribution with a Log Normal distribution, which assumes that there is 10% probability to be at −20% of the estimated CAPEX, and another 10% probability to be above 120% of the estimated CAPEX. These are based on observation of real cases, and

taken from literature, for plants that have been built, so at this early stage definition the capital cost estimation cannot be more precise than that.

In the big plant scenario (50,000 tons/year), the Lange and Petley estimation methods give an investment of 105 M US$ and 66 M US$, respectively.

When we extrapolate the historical (mostly acrylic acid) plant cost data with the power law method:

$$\frac{C_1}{C_2} = \left(\frac{S_1}{S_2}\right)^{0.65} \tag{4}$$

where $C$ is the cost (ISBL or OSBL, etc.) and $S$ the plant size, the scaled acrylic acid plant at 50,000 tons/year would therefore be in the 100 to 200 M US$ range. In acrylic acid plants, the two reactor costs can represent one-third of the total plant cost. Since the "Oxidative Coupling of Alcohols" has a single reactor, we considered it to be slightly cheaper than acrylic acid plants.

On the basis of these data, we assumed a fixed capital invested of 100 M US$. This translates to a CAPEX of 120 M US $ when including working capital (e.g., investment in 2 month of feedstocks/products or 4–7% of the capital invested), start-up cost (about 5%) and some contingencies as history shows that there are always some extra costs, with a fairly good confidence.

Similarly, for the 10,000 tons/year plant, the Lange and Petley methods estimated 28 M US$ and 33 M US $, respectively. The Lange method reaches the limits of the validity of the correlation, so it tends to underestimate the plant cost. The scaled-down investment based on expert judgement of literature data would be in the range of 35–75 M US$, to finally assume a fixed capital invested of 35 M US$, and a CAPEX of 42 M US$, Figure 7.

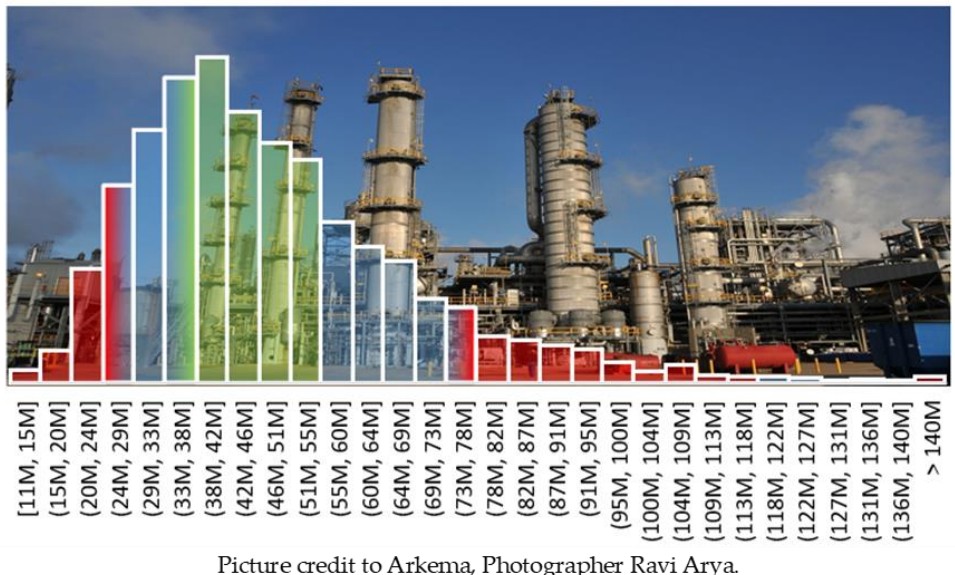

Capital Cost (ISBL+OSBL+Working Capital)
10 % to 90 % probability range: [27 M, 76 M]
Probabilities at Capex of 35 M US$ (27 %), Median Capex of 42 M US$, Median Capex -20 % (24 %) and at Median Capex + 30 % (68 %)
Log-Normal. Mean: 17.62, σ: 0.41

Picture credit to Arkema, Photographer Ravi Arya.

**Figure 7.** Capital cost distribution for a small plant (10 ktons/year)— CAPEX −20%/+ 30% (green area) corresponds to Class 3 confidence interval of the Association for the Advancement of Cost Engineering (AACE) [73].

The historical prices for methanol, ethanol and acrylic acid have been taken to model the variability of the prices, and to identify the statistical distributions that best fit historical data. Since acrolein is not a product that has a sufficient market to be listed and regularly

reported, the best proxy is acrylic acid since it is derived from acrolein, and obviously uses the same raw material. The same parameters that affect the production cost of acrylic acid should affect acrolein production cost. However, the acrolein marketed price is higher than acrylic acid, since the plants are smaller and fixed cost contributions are higher, so fluctuations of acrolein prices have a lower amplitude.

Some market prices for acrolein could be collected from the trans-border sales, from customs databases [74]. The period of interest is the 2013–2016 period, Figure 8, since at that time the price of crude oil varied from about 100 $/barrel to less than 40 $/barrel. It also reflects prices that we have seen lately. Many shipments, from China to India, for cumulated annual volumes of several 100 tons per year have been identified. Thus, the transfer prices reflect what would be relevant for a small consumer/producer in the range of 10,000 tons/year plant. It is clear that two different price scenarios can be considered. In a small plant scenario, with on-site production, acrolein prices should be benchmarked with a high import price for small shipments (Figure 9). In a large plant, with internal consumption where acrolein is an intermediate, the acrolein price should be benchmarked with a lower range of acrolein production cost.

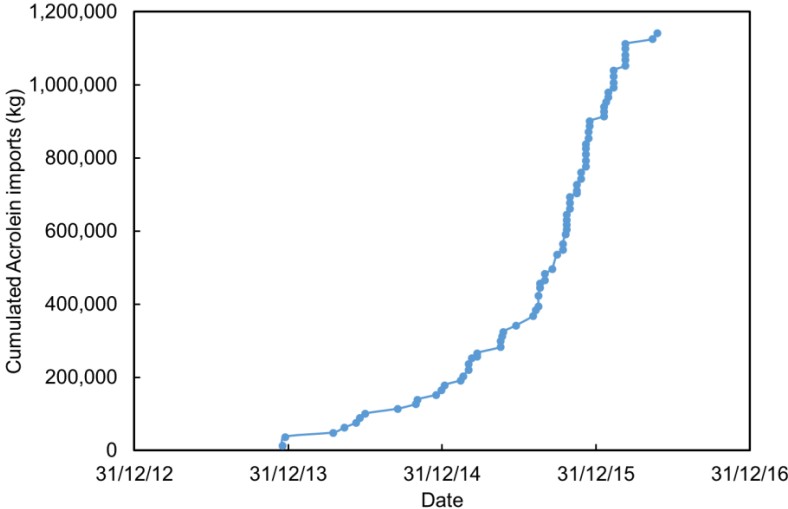

**Figure 8.** Cumulated quantities of acrolein imported in India between 2013 and 2016. Data computed from [74].

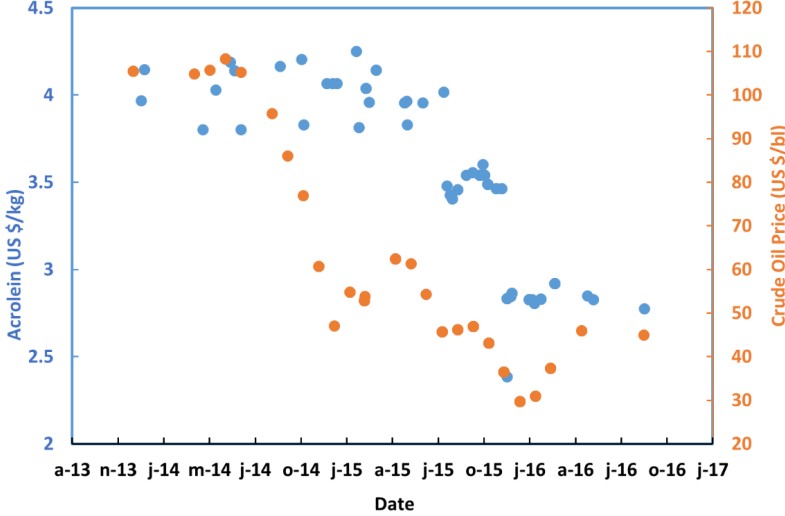

**Figure 9.** Small shipments acrolein prices [74] imported in India from China and crude oil price [75] trend between 2013 and 2016.

Prices are impacted similarly with the fluctuation of crude oil prices, with geo-climatic effects (like hurricanes, typhoons), which may disrupt petrochemical sites in the equatorial region, or serious drought such as those which affected the Rhine River and the petrochemical sites along it. The prices are also affected by unpredictable events like COVID19, Kuwait invasion, 11 September 2001, economic boom and the crisis that followed in 2008.

Although there is variability in the prices, they are not completely varying independently. For example, when the economy is booming, the demand for energy is rising, and people get better salary and expect to eat better (or more) food. Thus, we can see food prices increase along with energy prices. However, when there is an economic downturn, the energy prices would quickly drop, while food prices are going to remain high for some time. Thus, it is important to mimic the correlations in prices.

First, to give the best educated guess on the acrolein-feedstock correlation factor, we decided to correlate the acrolein price with the propylene and crude oil. Methanol and ethanol prices (Figure 10) are correlated with crude oil prices, which is also correlated with propylene, which in turn is correlated to acrolein. A correlation matrix was generated on the basis of historical prices (Table 3), but it was reviewed for all variables in order to assess if these past correlations would still apply in the future.

A new correlation matrix was then generated to represent our vision of the future (Table 4). Obviously, historical data of acrolein and acrylic acid differ, and the correlation matrix would have differed for small and big plants. However, although we considered two different acrolein distributions for small and big plants, due to the high level of uncertainties of the analysis, we used only one correlation matrix for both plants (Table 4).

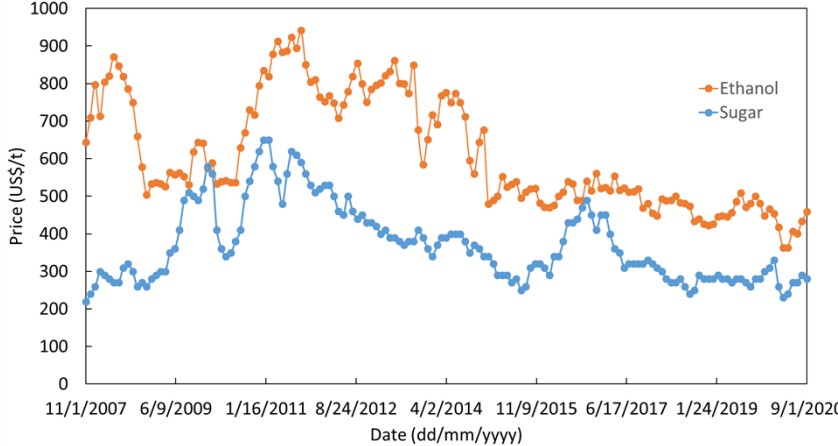

**Figure 10.** Ethanol [77] and sugar [76] prices in the US market between 2007 and 2020.

**Table 3.** Correlation matrix based on historical data, and considering acrolein price for a small plant.

|  | Ethanol | Methanol | Crude oil | Propylene | Acrolein |
|---|---|---|---|---|---|
| **Ethanol** | 1 |  |  |  |  |
| **Methanol** | 0.48 | 1 |  |  |  |
| **Crude oil** | 0.5 | 0.45 | 1 |  |  |
| **Propylene** | 0.45 | 0.45 | 0.5 | 1 |  |
| **Acrolein** | 0.45 | 0.5 | 0.45 | 0.5 | 1 |

**Table 4.** Proposed correlation matrix for the future, based on historical data and expert judgement.

|  | Ethanol | Methanol | Crude Oil | Propylene | Acrolein |
|---|---|---|---|---|---|
| **Ethanol** | 1 |  |  |  |  |
| **Methanol** | 0.7 | 1 |  |  |  |
| **Crude oil** | 0.5 | 0.45 | 1 |  |  |
| **Propylene** | 0.45 | 0.45 | 0.5 | 1 |  |
| **Acrolein** | 0.2 | 0.2 | 0.45 | 0.5 | 1 |

For all variables, a statistical distribution of prices had to be generated. Whether based on historical data, and keeping the same statistical laws, or based on expert judgment, the distribution can fit with normal, log-normal, gamma, Weibull or triangular distributions. The best fit for all the series is a log-normal distribution. We considered that in the future, methanol will become a more and more important feedstock, for instance due to the increasing interest of $CO_2$ hydrogenation, or $CH_4$ to methanol from natural gas or biogas. Its price will be somewhere around $300 \pm 100$ US\$/t (Figure 11). Ethanol, made by fermentation, has been historically related to the price of sugar [76,77] (Figure 10). When crude oil prices are low, ethylene hydration to ethanol plants becomes competitive again. Today, ethanol is also used as a price regulator for the sugar market, absorbing oscillation of the sugar production. For quite some time and until 2008/2009, ethanol prices in Europe were higher than in the US due to European regulations on sugar price. Additionally, after 2015, with regulations promoting lower consumption of soft drinks in US, the sugar consumption dropped and the price of ethanol started to decrease even more. We assumed that in the near future the price of ethanol will get closer to the US market prices, i.e., in the range of $500 \pm 150$ US\$/t, Figure 11.

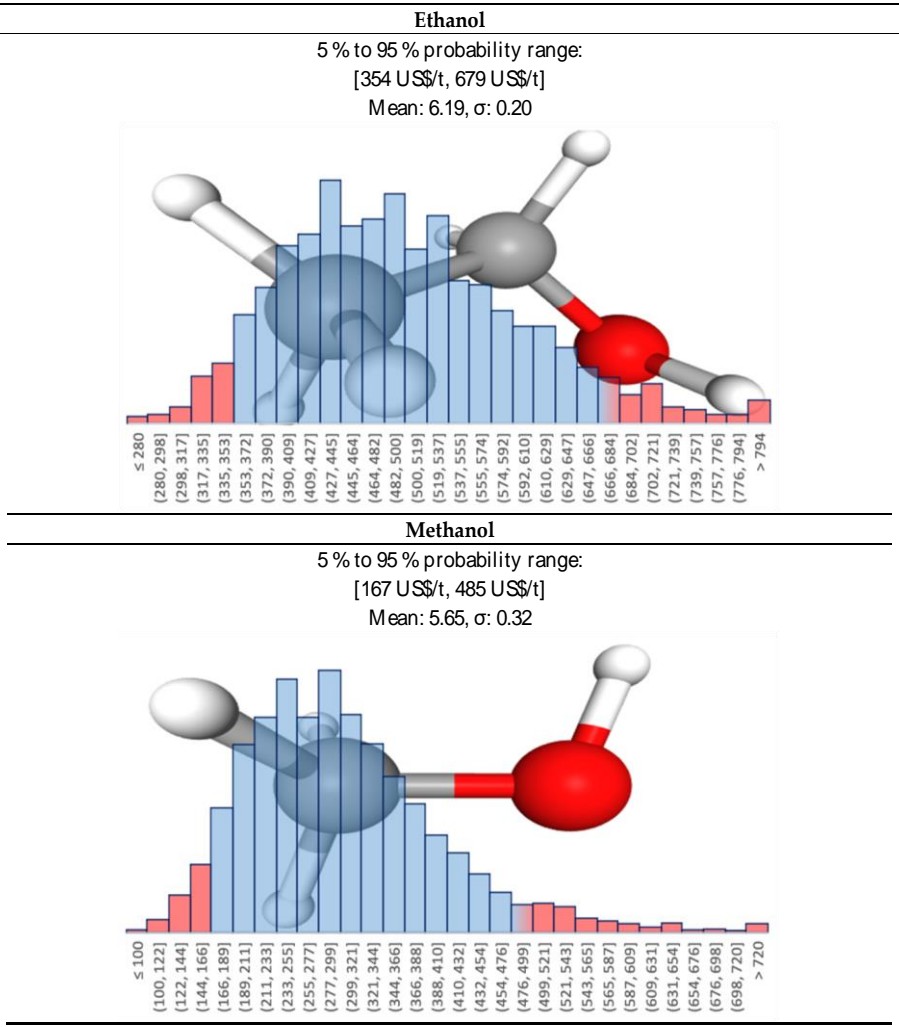

**Figure 11.** Ethanol and methanol individual price Log-Normal distributions used in the Monte-Carlo simulation.

Acrolein price needs to be differentiated between large and small demand because the consumers would be either internal or external. For a big plant, with internal consumption, we assumed that the marketed price would be in the range of 1800 to 3100 US\$/t, Figure 12.

**Acrolein**

Big Plant: 50 000 tons/year

5 % to 95 % probability range:
[1803 US$/t, 3108 US$/t]
Mean: 7.77, σ: 0.16

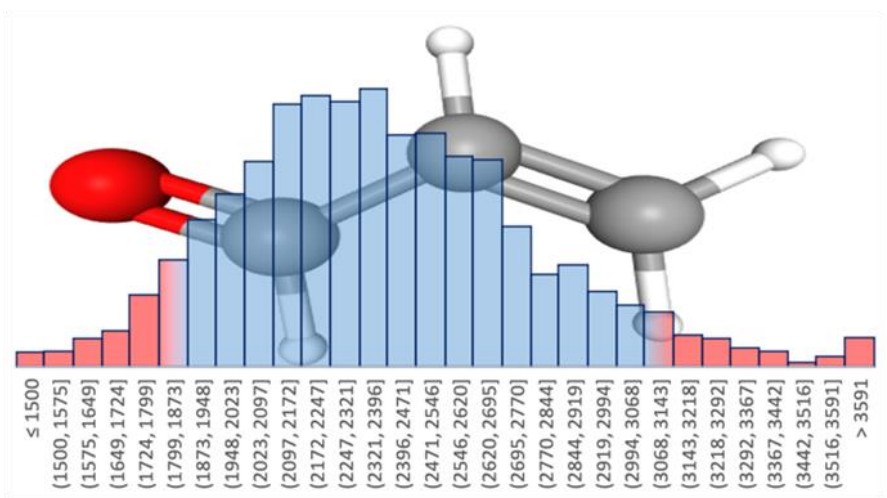

**Figure 12.** Acrolein price log-normal distributions for the big plant case.

Acrolein has a small open market; it is a very dangerous material to handle and store. Therefore, there are very few producers accepting to transport it, and it is difficult to guarantee a stable supply chain. A small acrolein plant would be ideally built close to or in the final consumers site, and therefore it would compete with the expensive few suppliers, often very far from the final consumer. At the same time, there would be less expenses for storage, transport (and insurance) while still offering a competitive price. For a small consumer, the transfer price would be higher, and we will be looking at 3900 to 5200 US$/t, that represents the spot price of acrolein for small tonnages (Figure 9) at a crude oil around 100 US$/barrel. The acrolein price distribution in the small plant case is reported in Figure 13.

The matrix calculated from the historical data is reported in Table 3. In the future, we foresee a higher correlation between bio-ethanol and methanol (0.7 vs. 0.48) because both are used to make biofuels, which are strongly impacted by subsidies and mandates. The independent statistical distributions are then correlated using the correlation matrix (Table 4), according to the methodology explained above and elsewhere [78]. Once correlated, the price distributions appear as illustrated in Figure 14.

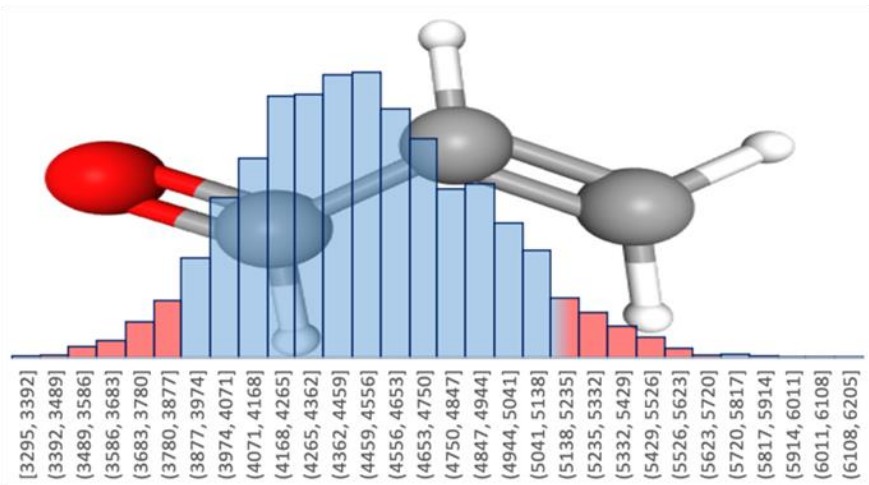

**Figure 13.** Acrolein prices Log-Normal distributions for small plant case.

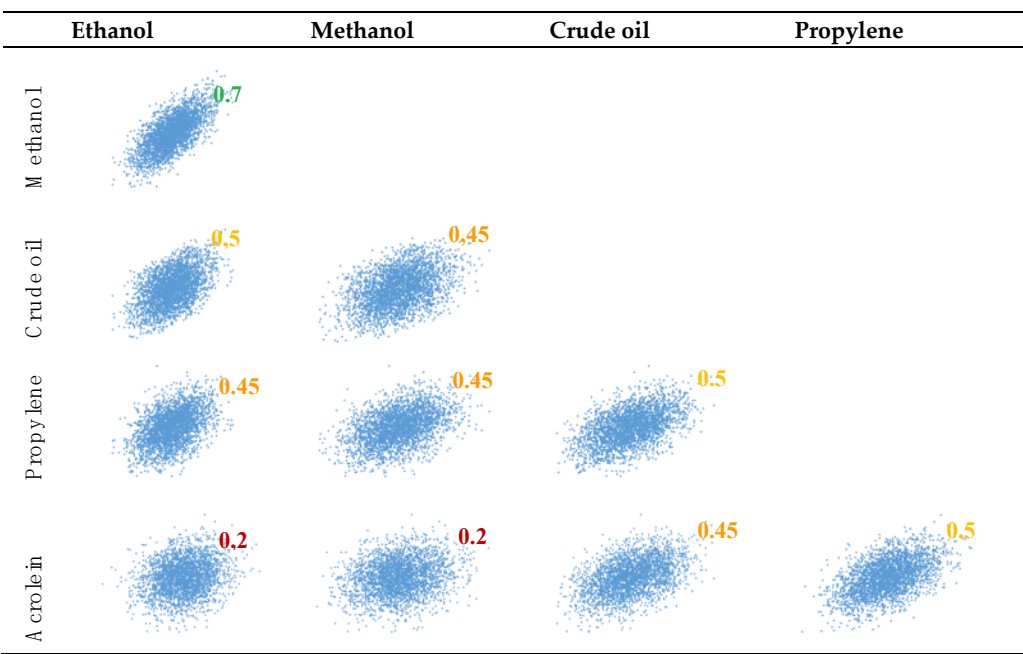

**Figure 14.** Visualization of the correlated prices of raw materials and products based on our vision of the future.

For acrolein, due to limited data, the historical correlations of acrolein/methanol and acrolein/ethanol for small and big plants are probably not relevant (0.2 and 0.001 for the big, and 0.5 and 0.45 for the small plants, respectively). For this reason, bridging through propylene and crude oil, we assumed an acrolein/methanol and acrolein/ethanol correlation coefficient of 0.2 in the future. The correlation matrix is reported in Table 4.

The next step is to calculate financial indicators. The plant is assumed to be built in France, where we benefit from a 35% tax rate, and built over 2 years, starting from year 0, with an increasing production rate (50% in year 2, 75% in year 3, 90% in year 4, and then 100% of the capacity for the following years). The internal rate of return (IRR) is taken as 10%.

The net present value (NPV) which is an indicator of the financial performance is calculated according to Perry 9th Edition [79]. The cumulated cash flow is calculated over the project duration.

The cumulated NPV is calculated for all the 3000 cases simulated annually, and then the probability to achieve each NPV is represented. The NPV is cumulated over 10 years of production (starting from year 0 to year 11, i.e., 2 years of construction and 10 years of production). The goal for all managements is to have a positive NPV, with a low probability of negative NPV (this is where a Monte-Carlo simulation is relevant). The Monte-Carlo simulation can cope with the high uncertainties of the plant capital cost, feedstock and product prices. It is used to identify the largest contributors to the cost of production, but also the largest contributors to the uncertainties.

Another goal is to cover the capital cost in a short period, in order to minimize the risks. Grants and subsidies on capital cost can indeed improve the economics, of a case that would make money on a longer term.

The model was used to simulate several cases, and this is where it gets all its value. The base cases are 10 and 50,000 tons/year, 70% yield.

When a new process is implemented, it is possible to obtain some grants and subsidies that could reduce the impact of the capital cost. In the different scenarios studied, we considered cases at different yields (70, 80 and 90%), but also with and without grants and subsidies (30% of the plant cost, which is an accessible value for a European plant). Figure 15 illustrates the capital cost distribution for the 50,000 tons/year scenario, with 30% subsidies. In fact, to be profitable (Figure 16), the big plant should realistically aim at a 70 M US$ capital cost, that would for example result from a 100 M$ investment subsidized at 30%. Considering 10 years of operation, at around 2 M$ annual labor cost, on top of 2 years of construction jobs, we believe 30 M$ of subsidies/grants to be achievable.

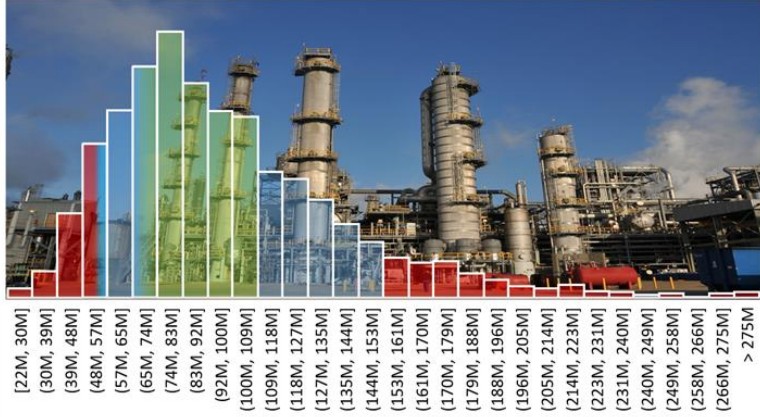

**Figure 15.** Capital cost distribution for a big plant (50 ktons/year) and taking into account 30% subsidies—CAPEX −20%/+ 30% (green area) corresponds to Class 3 confidence interval of the Association for the Advancement of Cost Engineering (AACE) [73].

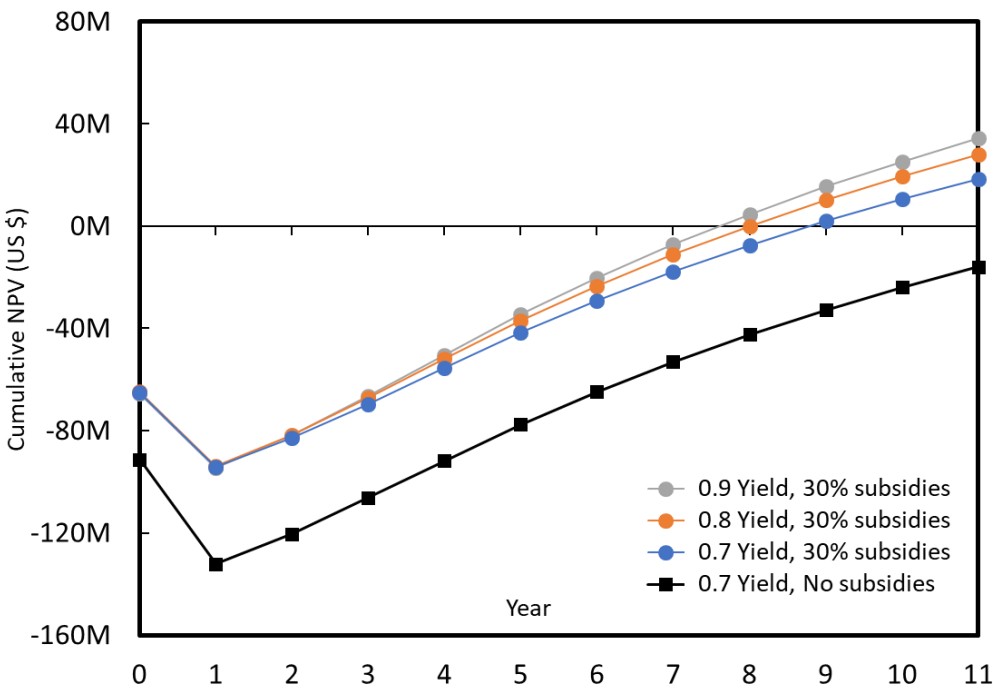

**Figure 16.** Cumulated NPV over 10 years of production, for a 50,000 tons/year plant, with and without 30% subsidies.

With the hypothesis made, without subsidies a large plant is unlikely to be implemented, as there is more than 50% probability to generate a negative NPV, after 10 years of production the plant capital cost is not yet covered. If 30% subsidies are obtained (or plant cost can be reduced with innovative new technologies), then the case is more likely, although it still takes 7 years to cover the plant cost, and it would be necessary to improve the model reliability (slope of the curve in Figure 17) as it will be illustrated by the tornado plot below. The impact of the increased yield is rather limited.

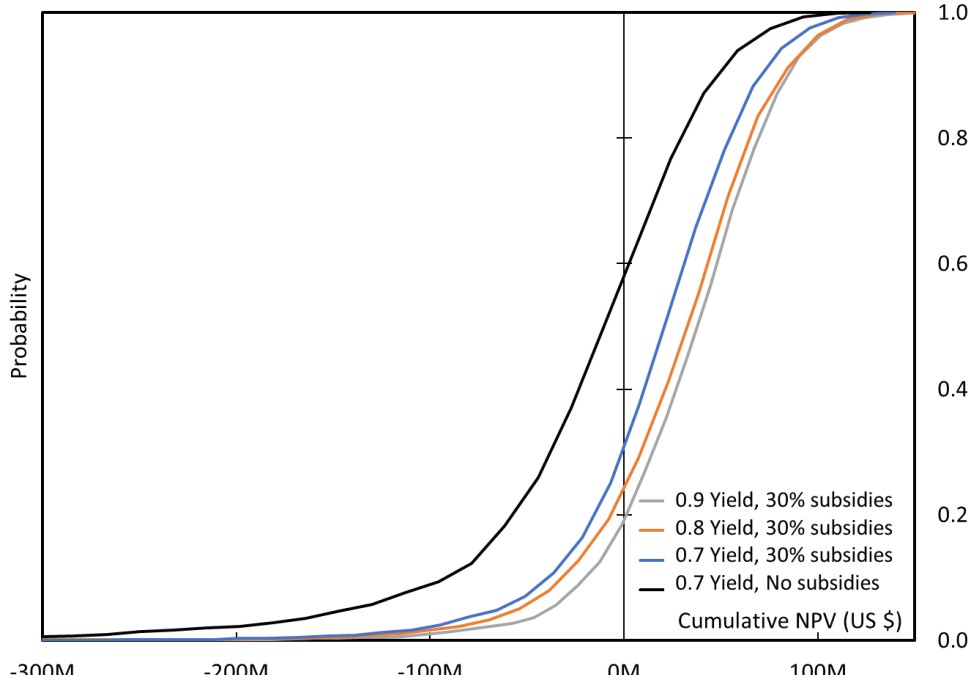

**Figure 17.** Probability vs. cumulated NPV in 10 years of production for a 50,000 tons/year plant with and without 30% subsidies.

On the 10,000 tons/year plant, the impact of the increased yield is also limited, but the impact of the grant or subsidies is significant. Even with a yield of only 70 mol %, the process would be likely implemented, although it still takes 7 years to cover the plant cost. If the acrolein price is more secured (lower dispersion), then the cumulated NPV is also more secured, and this would mean a higher slope on Figure 18, and a lower probability of negative NPV, Figure 19.

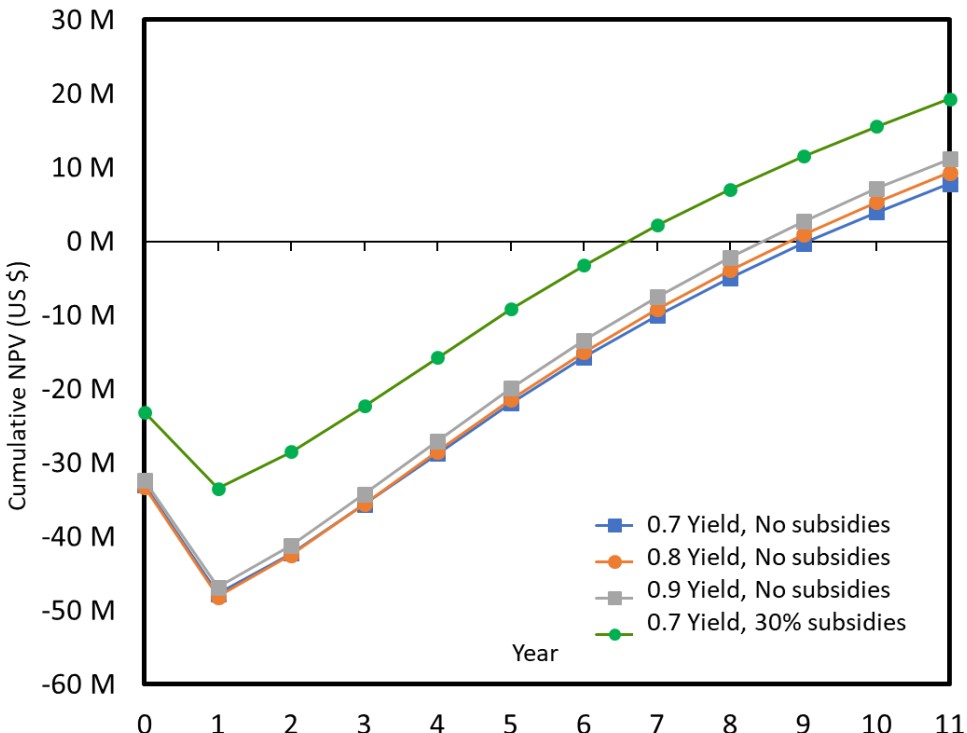

**Figure 18.** Cumulated NPV over 10 years of production for a 10,000 tons/year plant, with and without 30% subsidies.

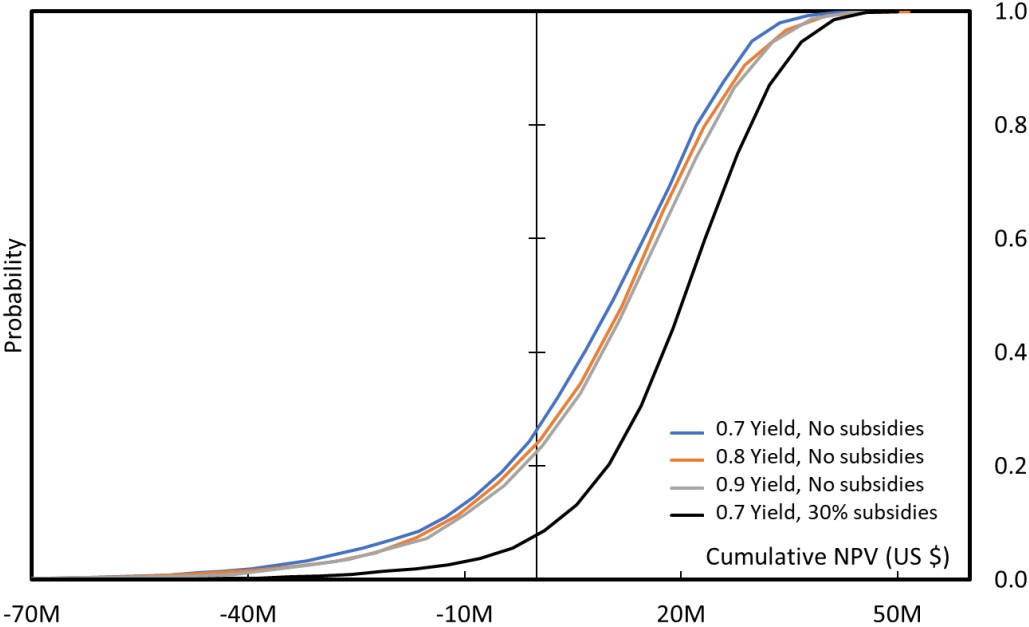

**Figure 19.** Probability of cumulated NPV over 10 years of production, for a 10,000 tons/year plant with and without 30% subsidies.

The tornado plot, Figure 20, illustrates the impact of the individual uncertainties (distributions) on the net present value (in the 10th year of production). It shows the impact of parameters at the extreme range (10 and 90% probabilities) of the statistical distributions. The parameters that have the highest impact are of course those for which it is important to improve the assumptions. For example, the impact on the NPV, of the methanol at its 10% or 90% probability values, is rather low compared with the impact of the other parameters. It also illustrates the impact of the distribution on the investment, on the NPV. In the case of the capital cost, it gives an indication to what extent it would be necessary to reduce it and make the case more favorable, or it can be used to look for the minimum plant size that would reduce the investment risk.

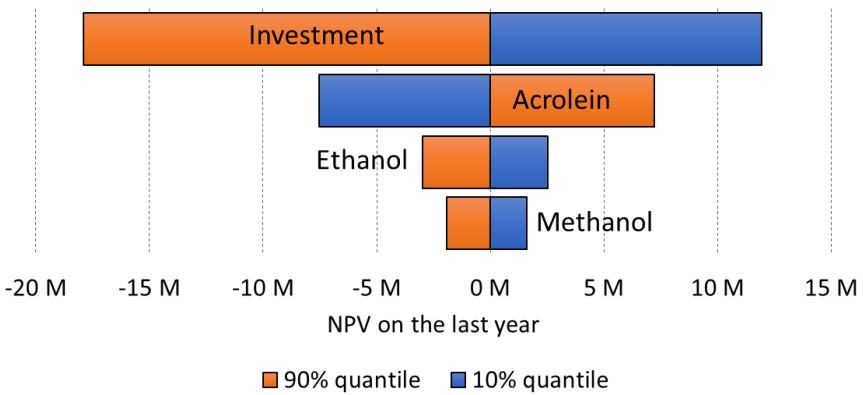

**Figure 20.** Tornado plot for a small plant (10,000 tons/year) at 70% yield, without subsidies. For the capital cost, we used a −20/+30% distribution on the median CAPEX corresponding to probability range of P24 and P68, respectively, and a Class 3 confidence.

We estimated the investment cost using early estimation methods (Lange and Petley), press releases and process review data. For the tornado plot, we decided to use the extreme values in the tornado at −20% and +30%, respectively, on the median CAPEX, that is the same level of confidence that a AACE (American Association Cost Engineers) class 3 would have, because here we can rely on similar plants which have been built.

Grants and subsidies reduce the investment cost, which in turn improves the chances to have a positive cumulated net present value. The uncertainties on the acrolein price (large distribution range) have almost as much impact on the NPV as the investment (Figure 21), so it would be wise to focus on securing a good deal with the final buyers of the company business unit.

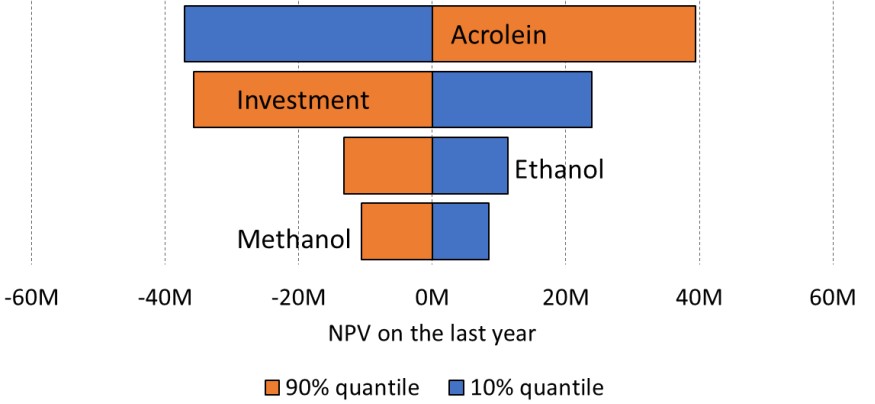

**Figure 21.** Tornado plot for a big plant (50,000 tons/year) at 70% yield with 30% subsidies on capital cost. For capital cost, we used a −20/+30% distribution on the median CAPEX corresponding to probability range of P24 and P68, respectively, and a Class 3 confidence.

For the large plant, the tornado plot shows that the uncertainty on the acrolein price has as much impact as the uncertainty on the plant capital cost. To improve the model reliability, it would be important to improve the confidence on the acrolein future price, but also on the definition of the process steps. It would also be important to keep the capital cost low. There are several options which have an impact on the type of catalysts to develop:

- the single reactor option was a very good choice and should be preserved;
- a simple technology should be thought to minimize the capital cost, so this would exclude circulating fluid beds, for example;
- side products should be minimized to avoid a heavy downstream purification;
- an existing plant could be retrofitted or second hand equipment could be used.

If a second hand multi-tubular reactor is selected, different reactor tube diameters and lengths are possible and would affect the choice of catalyst particle size.

As expected, the 10,000 tons/year plant has a higher median production cost than the 50,000 tons/year for the same parameters set (e.g., 2705 US$/ton vs. 1805 US$/ton for 70% yield without subsidies). The most important contributors to the production costs are the raw material costs, the other variable and fixed costs and the depreciation (Figure 22).

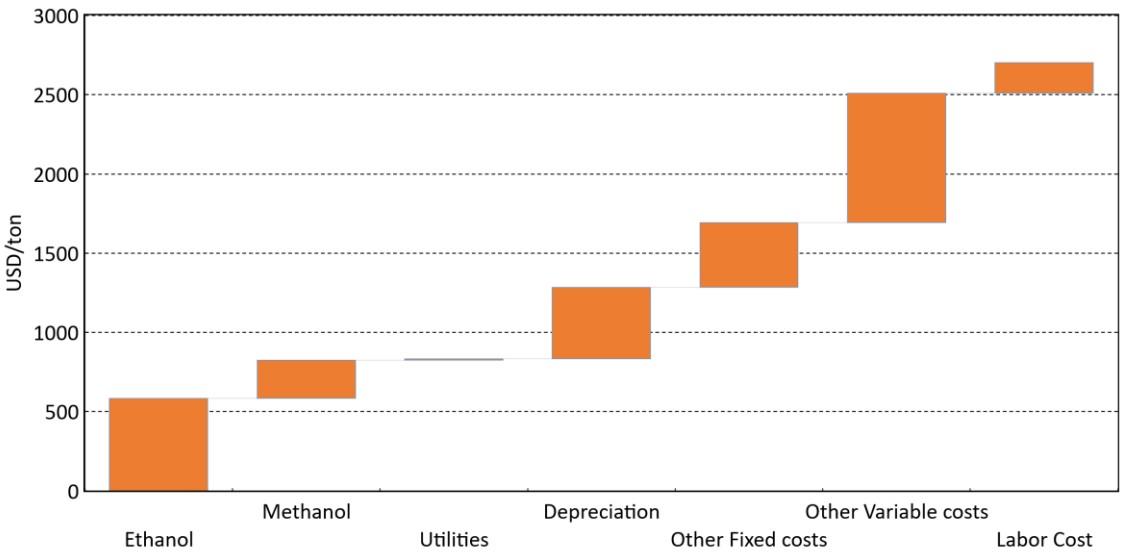

**Figure 22.** Production cost cascade for small plant (10,000 tons/year), 70% yield, without subsidies. Utilities (electricity and water) are calculated as 2% of raw materials cost. Depreciation is done over 10 years. Other fixed costs represent 9% of the capital cost. Other variable costs represent 18% of the sales. Labor cost represents 1.9 M US$/year.

Methanol, ethanol and utilities (on the left side of the graph) are directly linked with the catalyst efficiency (yield or selectivity). At higher yield, these contributions would be reduced. Utilities, here mostly electricity and water, were calculated as 2% of the raw material cost, since the process is self-sufficient in energy; there is little we can do to improve these contributors.

Depreciation is calculated on a flat rate over 10 years; and the other fixed costs (9% of the capital cost) are both directly connected to the process complexity. To reduce these contributors to the production cost, we need to keep the plant as simple as possible. It would be better to have less side products that would avoid recycling of unconverted streams. A single reactor is therefore a better choice.

Other variable costs are calculated as 18% of the sales, with 3% for royalties, 5% for R&D and 10% for distribution and sales. If acrolein is consumed internally, this contributor could be reduced. If the technology is developed internally, the R&D budget could be preserved, and the royalties reduced. Thus, the other variable costs, which are based on

the sales, unlike the labor cost, is the cost contribution that could be reduced to improve the economics.

On an operating plant, there is little potential to reduce the impact of the fixed costs, nor the depreciation. Usually savings are made on the other variable costs. In our scenario, we allocated 5% of the sales to R&D and 3% to royalties. There is then a lot of incentive to develop one's own technology and avoid paying royalties, but this might have to be done with a reduced R&D budget.

## 4. Remaining Challenges and Conclusions

Oxidative coupling of alcohols is a new reaction, which merits is to use widely available and potentially renewable feedstocks, in a single step and energy self-sufficient process to produce acrolein. Iron-molybdate combined with acid/base catalysts lead to acrolein, but other bifunctional oxidation catalysts should also be investigated. An appropriate acid/base balance is necessary as both sites are important to achieve a high selectivity. However, it is more important to characterize the equilibrated catalysts after a few hours of operation, while they are still active and selective.

The preliminary economic analysis shows that this route can compete with the classical propylene oxidation route, provided that the following conditions are met:

- equimolar methanol and ethanol feed;
- acrolein yield above 70 mol %—the analysis confirmed that 70% is a realistic target;
- single reactor, to minimize as much as possible the capital cost;
- 30% grants and subsidies on the capital cost, or a reduced capital cost with innovative technology.

The model can be used also to evaluate other scenarios, and it has been used in our early work to define in which conditions the catalysts should be tested, to have a chance to be able to up-scale it to commercial scale. In terms of catalyst development, we still need the following:

- to improve the aldolization reaction, which is slower than oxidation;
- reduce the over-oxidation to $CO_2$ and CO;
- lower the reaction temperature, while keeping it above 200 °C to satisfy the energy needs of the plant;
- evaluate the impact of the recirculation of unconverted aldehydes (formaldehyde and acetaldehyde) on the reaction kinetics and mechanism;
- develop single catalyst formulations, bulk or coated on inert support for better temperature control.

**Author Contributions:** Conceptualization, J.L.-D.; methodology, J.L.-D. and V.F.; software, J.L.-D. and J.d.T.; validation, V.F. and J.L.-D.; formal analysis, V.F., J.d.T. and J.L.-D.; investigation, J.L.-D. and V.F.; data curation, V.F., J.d.T., J.-L.D.; writing—original draft preparation, J.-L.D., V.F., J.d.T.; writing—review and editing, J.-L.D.; visualization, J.d.T.; supervision, J.-L.D. All authors have read and agreed to the published version of the manuscript.

**Funding:** This review received no external funding.

**Data Availability Statement:** Data available on request due to restrictions eg privacy or ethical. The data presented in this study are available on request from the corresponding author. The data are not publicly available due to privacy of the economic model.

**Acknowledgments:** Aline Auroux is acknowledged for her contribution to the research and her careful reading; contributions of A. Borowiec and A. Lilic former students (now graduated); M. Capron, S. Bennici. and G. Postole as supervisors; and Paul Masih and Antonin Balle as Jean-Luc Dubois's trainees students who contributed to build the economic model are deeply acknowledged.

**Conflicts of Interest:** Jean-Luc Dubois is the inventor of patents in references [23] and [24].

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
