# Peer review of "Review on Alternative Route to Acrolein through Oxidative Coupling of Alcohols"

_catalysts, doi:10.3390/catal11020229_

Round 1

Reviewer 1 Report

This manuscript focused on the acrolein production from methanol/ethanol oxidation and subsequent cross-aldolization of formaldehyde and acetaldehyde. The manuscript was well written and will be interest to readers in this field. Therefore, I recommend this manuscript to be published in your esteemed journal after minor revision. Minor comments 1. The authors compared methanol/ethanol route to propylene and glycerol routes for acrolein production. However, it is difficult to understand for readers, especially, in other fields. So, I suggest depicting the reaction pathway on these routes. 2. There are many mistakes (for example, Error! Reference source not found, may be EndNote software was used for organization of the reference) in the manuscript. The authors should carefully check it again.

Author Response

The authors thank the reviewer for his comments. 

Indeed there were many error messages generated from Hyperlinks when the paper was uploaded. The reason is unclear as it was not systematic. This has been corrected.

4 reactions have been explicitly added in "Scheme 1": the cross aldolization, the propylene oxidation, the glycerol dehydration and the oxidative coupling. This indeed should give a better view of the chemistries involved for those who are not familiar with them. Thanks a lot for requesting this clarification.

Best regards

Reviewer 2 Report

The authors have reviewed developments in the synthesis of acrolein and shown the advantage of the single step process and also the path how to commercialize the process. This review is useful not only for the readers dealing with catalysts or organic synthesis, but also the readers looking to commercialize their achievements of improved process with catalysts.

I think this work is acceptable after an appropriate revision.

My other comments are as follows:

There are many errors like “Error! Reference source not found”.

p.11 line 452

p.13 line 498

p.14 line 565

p.16 line 605

p.18 line 651

p.18 line 654

p.18 line 658

p.18 line 667

p.18 line 669

p.18 line 677

p.20 line 685

p.21 line 696

p.21 line 698

p.21 line 702

p.21 line 704

p.22 line 711

p.23 line 741

p.25 line 761

p.27 line 773

p.27 line 779

p.27 line 797

p.28 line 820

They may be fixed appropriately.

p.2 line 55 mutitubular

It may be fixed to multitubular.

p.7 line 291?

A citation for the No. 47 reference may not be found.

p.17 line 626 Figure 9

It may be fixed to Figure 8.

p.18 line 669 [67-68]

Please check those numbers in the citation.

Author Response

The authors thank the reviewer for his valuable comments. 

Indeed there were many error messages generated by hyperlinks when the paper was uploaded. The reason is unclear as it was not systematic. They have been corrected manually.

On page 2, MuLtitubular was corrected, along other misprints

On page 7, the citation 47 is now included in the block 46-49. There was a misprint previously. 

On page 17, the Reviewer is right, one should read "Figure 8".

On page 18 line 669 [67-68], references have been corrected (again it seems to be an Hyperlink issue)

Thanks again for all your comments.

Best regards

Reviewer 3 Report

This review rigorously studies the current processes for obtaining acrolein. The authors focus on the oxidative coupling of alcohols, where economic analysis is also extensively addressed. In section 3, Economic and Life Cycle Analysis, a very extensive and largely subjective economic analysis is performed for a chemical journal.

However, in order to improve the quality of the work, there are some points that, in my opinion, need to be modified.

  • It is necessary to specify and summarize section 3.
  • Figures 6, 7, 8, 9, 13, 15, 16, 17, 18 do not contribute to the compression of the article.
  • Numerous errors in lines: "Error! Reference source not found" 498, 565, 605, 651, 654,658, 667, 669, 677, 685, 696, 698, 703, 704, 711, 741, 742, 761, 773, 779, 820.

Author Response

The authors wants to thank the reviewer for his careful reading of the publications. 

There were indeed a lot of error messages generated by faulty hyperlinks when the publication was uploaded on the site. This has been corrected.

The Review paper was prepared with the intention to guide other researchers on what had been understood on the catalytic systems studied so far, but also to point to what still need to be done in order to successfuly implement the process at commercial scale. The Oxidative Coupling of Alcohols should also be seen as an example, that could be used in class rooms and lectures, on the analysis of problems to solve. The economic analysis is then an important part of the Review which has been prepared, in which we also give to the researcher tools and references where to find similar informations for other case studies. The methodology is on-purpose described extensively, so that people who are not expert in the field can be able to adapt it to their on case. It is also a goal in this review to guide future work in the right direction, and it is important that investigators know which catalyst improvements are valuable by identfying how it can improve the overall economics, and where to allocate most of their efforts. 

For those who would like to directly jump to the summary and the conclusions, they will find them in Section 4, were the key improvements still needed in terms of catalysts and/or process are listed and explained. In that sense, Section 3 is an essential part of the review, and we advise to keep it in the manuscript rather than to move it into a "supplementary information" section.

Best regards